# Combinatorial Multi-Armed Bandit with General Reward Functions

Wei Chen[*]      Wei Hu[†]      Fu Li[‡]      Jian Li[§]      Yu Liu[¶]      Pinyan Lu[‖]

## Abstract

In this paper, we study the stochastic combinatorial multi-armed bandit (CMAB) framework that allows a general nonlinear reward function, whose expected value may not depend only on the means of the input random variables but possibly on the entire distributions of these variables. Our framework enables a much larger class of reward functions such as the $\max()$ function and nonlinear utility functions. Existing techniques relying on accurate estimations of the means of random variables, such as the upper confidence bound (UCB) technique, do not work directly on these functions. We propose a new algorithm called *stochastically dominant confidence bound (SDCB)*, which estimates the distributions of underlying random variables and their stochastically dominant confidence bounds. We prove that SDCB can achieve $O(\log T)$ distribution-dependent regret and $\tilde{O}(\sqrt{T})$ distribution-independent regret, where $T$ is the time horizon. We apply our results to the $K$-MAX problem and expected utility maximization problems. In particular, for $K$-MAX, we provide the first polynomial-time approximation scheme (PTAS) for its offline problem, and give the first $\tilde{O}(\sqrt{T})$ bound on the $(1-\epsilon)$-approximation regret of its online problem, for any $\epsilon > 0$.

## 1 Introduction

Stochastic multi-armed bandit (MAB) is a classical online learning problem typically specified as a player against $m$ machines or arms. Each arm, when pulled, generates a random reward following an unknown distribution. The task of the player is to select one arm to pull in each round based on the historical rewards she collected, and the goal is to collect cumulative reward over multiple rounds as much as possible. In this paper, unless otherwise specified, we use MAB to refer to stochastic MAB.

MAB problem demonstrates the key tradeoff between exploration and exploitation: whether the player should stick to the choice that performs the best so far, or should try some less explored alternatives that may provide better rewards. The performance measure of an MAB strategy is its cumulative *regret*, which is defined as the difference between the cumulative reward obtained by always playing the arm with the largest expected reward and the cumulative reward achieved by the learning strategy. MAB and its variants have been extensively studied in the literature, with classical results such as tight $\Theta(\log T)$ distribution-dependent and $\Theta(\sqrt{T})$ distribution-independent upper and lower bounds on the regret in $T$ rounds [19, 2, 1].

An important extension to the classical MAB problem is combinatorial multi-armed bandit (CMAB). In CMAB, the player selects not just one arm in each round, but a subset of arms or a combinatorial

---

[*]Microsoft Research, email: `weic@microsoft.com`. The authors are listed in alphabetical order.

[†]Princeton University, email: `huwei@cs.princeton.edu`.

[‡]The University of Texas at Austin, email: `fuli.theory.research@gmail.com`.

[§]Tsinghua University, email: `lapordge@gmail.com`.

[¶]Tsinghua University, email: `liuyujyyz@gmail.com`.

[‖]Shanghai University of Finance and Economics, email: `lu.pinyan@mail.shufe.edu.cn`.

object in general, referred to as a super arm, which collectively provides a random reward to the player. The reward depends on the outcomes from the selected arms. The player may observe partial feedbacks from the selected arms to help her in decision making. CMAB has wide applications in online advertising, online recommendation, wireless routing, dynamic channel allocations, etc., because in all these settings the action unit is a combinatorial object (e.g. a set of advertisements, a set of recommended items, a route in a wireless network, and an allocation between channels and users), and the reward depends on unknown stochastic behaviors (e.g. users' click through behaviors, wireless transmission quality, etc.). Therefore CMAB has attracted a lot of attention in online learning research in recent years [12, 8, 22, 15, 7, 16, 18, 17, 23, 9].

Most of these studies focus on linear reward functions, for which the expected reward for playing a super arm is a linear combination of the expected outcomes from the constituent base arms. Even for studies that do generalize to non-linear reward functions, they typically still assume that the expected reward for choosing a super arm is a function of the expected outcomes from the constituent base arms in this super arm [8, 17]. However, many natural reward functions do not satisfy this property. For example, for the function $\max()$, which takes a group of variables and outputs the maximum one among them, its expectation depends on the full distributions of the input random variables, not just their means. Function $\max()$ and its variants underly many applications. As an illustrative example, we consider the following scenario in auctions: the auctioneer is repeatedly selling an item to $m$ bidders; in each round the auctioneer selects $K$ bidders to bid; each of the $K$ bidders independently draws her bid from her private valuation distribution and submits the bid; the auctioneer uses the first-price auction to determine the winner and collects the largest bid as the payment.[1] The goal of the auctioneer is to gain as high cumulative payments as possible. We refer to this problem as the $K$-MAX bandit problem, which cannot be effectively solved in the existing CMAB framework.

Beyond the $K$-MAX problem, many expected utility maximization (EUM) problems are studied in stochastic optimization literature [27, 20, 21, 4]. The problem can be formulated as maximizing $\mathbb{E}[u(\sum_{i \in S} X_i)]$ among all feasible sets $S$, where $X_i$'s are independent random variables and $u(\cdot)$ is a utility function. For example, $X_i$ could be the random delay of edge $e_i$ in a routing graph, $S$ is a routing path in the graph, and the objective is maximizing the utility obtained from any routing path, and typically the shorter the delay, the larger the utility. The utility function $u(\cdot)$ is typically nonlinear to model risk-averse or risk-prone behaviors of users (e.g. a concave utility function is often used to model risk-averse behaviors). The non-linear utility function makes the objective function much more complicated: in particular, it is no longer a function of the means of the underlying random variables $X_i$'s. When the distributions of $X_i$'s are unknown, we can turn EUM into an online learning problem where the distributions of $X_i$'s need to be learned over time from online feedbacks, and we want to maximize the cumulative reward in the learning process. Again, this is not covered by the existing CMAB framework since only learning the means of $X_i$'s is not enough.

In this paper, we generalize the existing CMAB framework with semi-bandit feedbacks to handle general reward functions, where the expected reward for playing a super arm may depend more than just the means of the base arms, and the outcome distribution of a base arm can be arbitrary. This generalization is non-trivial, because almost all previous works on CMAB rely on estimating the expected outcomes from base arms, while in our case, we need an estimation method and an analytical tool to deal with the whole distribution, not just its mean. To this end, we turn the problem into estimating the cumulative distribution function (CDF) of each arm's outcome distribution. We use *stochastically dominant confidence bound (SDCB)* to obtain a distribution that stochastically dominates the true distribution with high probability, and hence we also name our algorithm SDCB. We are able to show $O(\log T)$ distribution-dependent and $\tilde{O}(\sqrt{T})$ distribution-independent regret bounds in $T$ rounds. Furthermore, we propose a more efficient algorithm called Lazy-SDCB, which first executes a discretization step and then applies SDCB on the discretized problem. We show that Lazy-SDCB also achieves $\tilde{O}(\sqrt{T})$ distribution-independent regret bound. Our regret bounds are tight with respect to their dependencies on $T$ (up to a logarithmic factor for distribution-independent bounds). To make our scheme work, we make a few reasonable assumptions, including boundedness, monotonicity and Lipschitz-continuity[2] of the reward function, and independence among base arms. We apply our algorithms to the $K$-MAX and EUM problems, and provide efficient solutions with concrete regret bounds. Along the way, we also provide the first polynomial time approximation

scheme (PTAS) for the offline $K$-MAX problem, which is formulated as maximizing $\mathbb{E}[\max_{i \in S} X_i]$ subject to a cardinality constraint $|S| \leq K$, where $X_i$'s are independent nonnegative random variables.

To summarize, our contributions include: (a) generalizing the CMAB framework to allow a general reward function whose expectation may depend on the entire distributions of the input random variables; (b) proposing the SDCB algorithm to achieve efficient learning in this framework with near-optimal regret bounds, even for arbitrary outcome distributions; (c) giving the first PTAS for the offline $K$-MAX problem. Our general framework treats any offline stochastic optimization algorithm as an oracle, and effectively integrates it into the online learning framework.

**Related Work.** As already mentioned, most relevant to our work are studies on CMAB frameworks, among which [12, 16, 18, 9] focus on linear reward functions while [8, 17] look into non-linear reward functions. In particular, Chen et al. [8] look at general non-linear reward functions and Kveton et al. [17] consider specific non-linear reward functions in a conjunctive or disjunctive form, but both papers require that the expected reward of playing a super arm is determined by the expected outcomes from base arms.

The only work in combinatorial bandits we are aware of that does not require the above assumption on the expected reward is [15], which is based on a general Thompson sampling framework. However, they assume that the joint distribution of base arm outcomes is from a known parametric family within known likelihood function and only the parameters are unknown. They also assume the parameter space to be finite. In contrast, our general case is non-parametric, where we allow arbitrary bounded distributions. Although in our known finite support case the distribution can be parametrized by probabilities on all supported points, our parameter space is continuous. Moreover, it is unclear how to efficiently compute posteriors in their algorithm, and their regret bounds depend on complicated problem-dependent coefficients which may be very large for many combinatorial problems. They also provide a result on the $K$-MAX problem, but they only consider Bernoulli outcomes from base arms, much simpler than our case where general distributions are allowed.

There are extensive studies on the classical MAB problem, for which we refer to a survey by Bubeck and Cesa-Bianchi [5]. There are also some studies on adversarial combinatorial bandits, e.g. [26, 6]. Although it bears conceptual similarities with stochastic CMAB, the techniques used are different.

Expected utility maximization (EUM) encompasses a large class of stochastic optimization problems and has been well studied (e.g. [27, 20, 21, 4]). To the best of our knowledge, we are the first to study the online learning version of these problems, and we provide a general solution to systematically address all these problems as long as there is an available offline (approximation) algorithm. The $K$-MAX problem may be traced back to [13], where Goel et al. provide a constant approximation algorithm to a generalized version in which the objective is to choose a subset $S$ of cost at most $K$ and maximize the expectation of a certain knapsack profit.

## 2 Setup and Notation

**Problem Formulation.** We model a combinatorial multi-armed bandit (CMAB) problem as a tuple $(E, \mathcal{F}, D, R)$, where $E = [m] = \{1, 2, \ldots, m\}$ is a set of $m$ (base) arms, $\mathcal{F} \subseteq 2^E$ is a set of subsets of $E$, $D$ is a probability distribution over $[0, 1]^m$, and $R$ is a reward function defined on $[0, 1]^m \times \mathcal{F}$. The arms produce stochastic outcomes $X = (X_1, X_2, \ldots, X_m)$ drawn from distribution $D$, where the $i$-th entry $X_i$ is the outcome from the $i$-th arm. Each feasible subset of arms $S \in \mathcal{F}$ is called a *super arm*. Under a realization of outcomes $x = (x_1, \ldots, x_m)$, the player receives a reward $R(x, S)$ when she chooses the super arm $S$ to play. Without loss of generality, we assume the reward value to be nonnegative. Let $K = \max_{S \in \mathcal{F}} |S|$ be the maximum size of any super arm.

Let $X^{(1)}, X^{(2)}, \ldots$ be an i.i.d. sequence of random vectors drawn from $D$, where $X^{(t)} = (X_1^{(t)}, \ldots, X_m^{(t)})$ is the outcome vector generated in the $t$-th round. In the $t$-th round, the player chooses a super arm $S_t \in \mathcal{F}$ to play, and then the outcomes from all arms in $S_t$, i.e., $\{X_i^{(t)} \mid i \in S_t\}$, are revealed to the player. According to the definition of the reward function, the reward value in the $t$-th round is $R(X^{(t)}, S_t)$. The expected reward for choosing a super arm $S$ in any round is denoted by $r_D(S) = \mathbb{E}_{X \sim D}[R(X, S)]$.

We also assume that for a fixed super arm $S \in \mathcal{F}$, the reward $R(x, S)$ only depends on the revealed outcomes $x_S = (x_i)_{i \in S}$. Therefore, we can alternatively express $R(x, S)$ as $R_S(x_S)$, where $R_S$ is a function defined on $[0, 1]^S$.[3]

A learning algorithm $\mathcal{A}$ for the CMAB problem selects which super arm to play in each round based on the revealed outcomes in all previous rounds. Let $S_t^{\mathcal{A}}$ be the super arm selected by $\mathcal{A}$ in the $t$-th round.[4] The goal is to maximize the expected cumulative reward in $T$ rounds, which is $\mathbb{E}\left[\sum_{t=1}^{T} R(X^{(t)}, S_t^{\mathcal{A}})\right] = \sum_{t=1}^{T} \mathbb{E}\left[r_D(S_t^{\mathcal{A}})\right]$. Note that when the underlying distribution $D$ is known, the optimal algorithm $\mathcal{A}^*$ chooses the optimal super arm $S^* = \mathrm{argmax}_{S \in \mathcal{F}}\{r_D(S)\}$ in every round. The quality of an algorithm $\mathcal{A}$ is measured by its *regret* in $T$ rounds, which is the difference between the expected cumulative reward of the optimal algorithm $\mathcal{A}^*$ and that of $\mathcal{A}$:

$$\mathrm{Reg}_D^{\mathcal{A}}(T) = T \cdot r_D(S^*) - \sum_{t=1}^{T} \mathbb{E}\left[r_D(S_t^{\mathcal{A}})\right].$$

For some CMAB problem instances, the optimal super arm $S^*$ may be computationally hard to find even when the distribution $D$ is known, but efficient approximation algorithms may exist, i.e., an $\alpha$-approximate ($0 < \alpha \leq 1$) solution $S' \in \mathcal{F}$ which satisfies $r_D(S') \geq \alpha \cdot \max_{S \in \mathcal{F}}\{r_D(S)\}$ can be efficiently found given $D$ as input. We will provide the exact formulation of our requirement on such an $\alpha$-*approximation computation oracle* shortly. In such cases, it is not fair to compare a CMAB algorithm $\mathcal{A}$ with the optimal algorithm $\mathcal{A}^*$ which always chooses the optimal super arm $S^*$. Instead, we define the $\alpha$-*approximation regret* of an algorithm $\mathcal{A}$ as

$$\mathrm{Reg}_{D,\alpha}^{\mathcal{A}}(T) = T \cdot \alpha \cdot r_D(S^*) - \sum_{t=1}^{T} \mathbb{E}\left[r_D(S_t^{\mathcal{A}})\right].$$

As mentioned, almost all previous work on CMAB requires that the expected reward $r_D(S)$ of a super arm $S$ depends only on the expectation vector $\mu = (\mu_1, \ldots, \mu_m)$ of outcomes, where $\mu_i = \mathbb{E}_{X \sim D}[X_i]$. This is a strong restriction that cannot be satisfied by a general nonlinear function $R_S$ and a general distribution $D$. The main motivation of this work is to remove this restriction.

**Assumptions.** Throughout this paper, we make several assumptions on the outcome distribution $D$ and the reward function $R$.

**Assumption 1** (Independent outcomes from arms). *The outcomes from all $m$ arms are mutually independent, i.e., for $X \sim D$, $X_1, X_2, \ldots, X_m$ are mutually independent. We write $D$ as $D = D_1 \times D_2 \times \cdots \times D_m$, where $D_i$ is the distribution of $X_i$.*

We remark that the above independence assumption is also made for past studies on the offline EUM and K-MAX problems [27, 20, 21, 4, 13], so it is not an extra assumption for the online learning case.

**Assumption 2** (Bounded reward value). *There exists $M > 0$ such that for any $x \in [0, 1]^m$ and any $S \in \mathcal{F}$, we have $0 \leq R(x, S) \leq M$.*

**Assumption 3** (Monotone reward function). *If two vectors $x, x' \in [0, 1]^m$ satisfy $x_i \leq x_i'$ ($\forall i \in [m]$), then for any $S \in \mathcal{F}$, we have $R(x, S) \leq R(x', S)$.*

**Computation Oracle for Discrete Distributions with Finite Supports.** We require that there exists an $\alpha$-approximation computation oracle ($0 < \alpha \leq 1$) for maximizing $r_D(S)$, when each $D_i$ ($i \in [m]$) has a *finite support*. In this case, $D_i$ can be fully described by a finite set of numbers (i.e., its support $\{v_{i,1}, v_{i,2}, \ldots, v_{i,s_i}\}$ and the values of its cumulative distribution function (CDF) $F_i$ on the supported points: $F_i(v_{i,j}) = \mathrm{Pr}_{X_i \sim D_i}[X_i \leq v_{i,j}]$ ($j \in [s_i]$)). The oracle takes such a representation of $D$ as input, and can output a super arm $S' = \mathrm{Oracle}(D) \in \mathcal{F}$ such that $r_D(S') \geq \alpha \cdot \max_{S \in \mathcal{F}}\{r_D(S)\}$.

## 3 SDCB Algorithm

**Algorithm 1** SDCB (Stochastically dominant confidence bound)

---

1: Throughout the algorithm, for each arm $i \in [m]$, maintain: (i) a counter $T_i$ which stores the number of times arm $i$ has been played so far, and (ii) the empirical distribution $\hat{D}_i$ of the observed outcomes from arm $i$ so far, which is represented by its CDF $\hat{F}_i$

2: // Initialization
3: **for** $i = 1$ **to** $m$ **do**
4:      // Action in the $i$-th round
5:      Play a super arm $S_i$ that contains arm $i$
6:      Update $T_j$ and $\hat{F}_j$ for each $j \in S_i$
7: **end for**

8: **for** $t = m + 1, m + 2, \ldots$ **do**
9:      // Action in the $t$-th round
10:      For each $i \in [m]$, let $\underline{D}_i$ be a distribution whose CDF $\underline{F}_i$ is

$$\underline{F}_i(x) = \begin{cases} \max\{\hat{F}_i(x) - \sqrt{\frac{3\ln t}{2T_i}}, 0\}, & 0 \leq x < 1 \\ 1, & x = 1 \end{cases}$$

11:      Play the super arm $S_t \leftarrow \mathsf{Oracle}(\underline{D})$, where $\underline{D} = \underline{D}_1 \times \underline{D}_2 \times \cdots \times \underline{D}_m$
12:      Update $T_j$ and $\hat{F}_j$ for each $j \in S_t$
13: **end for**

---

We present our algorithm *stochastically dominant confidence bound (SDCB)* in Algorithm 1. Throughout the algorithm, we store, in a variable $T_i$, the number of times the outcomes from arm $i$ are observed so far. We also maintain the empirical distribution $\hat{D}_i$ of the observed outcomes from arm $i$ so far, which can be represented by its CDF $\hat{F}_i$: for $x \in [0, 1]$, the value of $\hat{F}_i(x)$ is just the fraction of the observed outcomes from arm $i$ that are no larger than $x$. Note that $\hat{F}_i$ is always a step function which has "jumps" at the points that are observed outcomes from arm $i$. Therefore it suffices to store these discrete points as well as the values of $\hat{F}_i$ at these points in order to store the whole function $\hat{F}_i$. Similarly, the later computation of stochastically dominant CDF $\underline{F}_i$ (line 10) only requires computation at these points, and the input to the offline oracle only needs to provide these points and corresponding CDF values (line 11).

The algorithm starts with $m$ initialization rounds in which each arm is played at least once[5] (lines 2-7). In the $t$-th round ($t > m$), the algorithm consists of three steps. First, it calculates for each $i \in [m]$ a distribution $\underline{D}_i$ whose CDF $\underline{F}_i$ is obtained by lowering the CDF $\hat{F}_i$ (line 10). The second step is to call the $\alpha$-approximation oracle with the newly constructed distribution $\underline{D} = \underline{D}_1 \times \cdots \times \underline{D}_m$ as input (line 11), and thus the super arm $S_t$ output by the oracle satisfies $r_{\underline{D}}(S_t) \geq \alpha \cdot \max_{S \in \mathcal{F}}\{r_{\underline{D}}(S)\}$. Finally, the algorithm chooses the super arm $S_t$ to play, observes the outcomes from all arms in $S_t$, and updates $T_j$'s and $\hat{F}_j$'s accordingly for each $j \in S_t$.

The idea behind our algorithm is the *optimism in the face of uncertainty* principle, which is the key principle behind UCB-type algorithms. Our algorithm ensures that with high probability we have $\underline{F}_i(x) \leq F_i(x)$ simultaneously for all $i \in [m]$ and all $x \in [0, 1]$, where $F_i$ is the CDF of the outcome distribution $D_i$. This means that each $\underline{D}_i$ has *first-order stochastic dominance* over $D_i$.[6] Then from the monotonicity property of $R(x, S)$ (Assumption 3) we know that $r_{\underline{D}}(S) \geq r_D(S)$ holds for all $S \in \mathcal{F}$ with high probability. Therefore $\underline{D}$ provides an "optimistic" estimation on the expected reward from each super arm.

**Regret Bounds.** We prove $O(\log T)$ distribution-dependent and $O(\sqrt{T \log T})$ distribution-independent upper bounds on the regret of SDCB (Algorithm 1).

We call a super arm $S$ *bad* if $r_D(S) < \alpha \cdot r_D(S^*)$. For each super arm $S$, we define

$$\Delta_S = \max\{\alpha \cdot r_D(S^*) - r_D(S), 0\}.$$

Let $\mathcal{F}_{\mathrm{B}} = \{S \in \mathcal{F} \mid \Delta_S > 0\}$, which is the set of all *bad* super arms. Let $E_{\mathrm{B}} \subseteq [m]$ be the set of arms that are contained in at least one *bad* super arm. For each $i \in E_{\mathrm{B}}$, we define

$$\Delta_{i,\min} = \min\{\Delta_S \mid S \in \mathcal{F}_{\mathrm{B}}, i \in S\}.$$

Recall that $M$ is an upper bound on the reward value (Assumption 2) and $K = \max_{S \in \mathcal{F}} |S|$.

**Theorem 1.** *A distribution-dependent upper bound on the $\alpha$-approximation regret of* SDCB *(Algorithm 1) in $T$ rounds is*

$$M^2 K \sum_{i \in E_{\mathrm{B}}} \frac{2136}{\Delta_{i,\min}} \ln T + \left(\frac{\pi^2}{3} + 1\right) \alpha M m,$$

*and a distribution-independent upper bound is*

$$93 M \sqrt{mKT \ln T} + \left(\frac{\pi^2}{3} + 1\right) \alpha M m.$$

The proof of Theorem 1 is given in the supplementary material. The main idea is to reduce our analysis on general reward functions satisfying Assumptions 1-3 to the one in [18] that deals with the *summation* reward function $R(x, S) = \sum_{i \in S} x_i$. Our analysis relies on the Dvoretzky-Kiefer-Wolfowitz inequality [10, 24], which gives a uniform concentration bound on the empirical CDF of a distribution.

**Applying Our Algorithm to the Previous CMAB Framework.** Although our focus is on general reward functions, we note that when SDCB is applied to the previous CMAB framework where the expected reward depends only on the means of the random variables, it can achieve the same regret bounds as the previous *combinatorial upper confidence bound (*CUCB*)* algorithm in [8, 18].

Let $\mu_i = \mathbb{E}_{X \sim D}[X_i]$ be arm $i$'s mean outcome. In each round CUCB calculates (for each arm $i$) an upper confidence bound $\bar{\mu}_i$ on $\mu_i$, with the essential property that $\mu_i \le \bar{\mu}_i \le \mu_i + \Lambda_i$ holds with high probability, for some $\Lambda_i > 0$. In SDCB, we use $\underline{D}_i$ as a stochastically dominant confidence bound of $D_i$. We can show that $\mu_i \le \mathbb{E}_{Y_i \sim \underline{D}_i}[Y_i] \le \mu_i + \Lambda_i$ holds with high probability, with the same interval length $\Lambda_i$ as in CUCB. (The proof is given in the supplementary material.) Hence, the analysis in [8, 18] can be applied to SDCB, resulting in the same regret bounds. We further remark that in this case we *do not* need the three assumptions stated in Section 2 (in particular the independence assumption on $X_i$'s): the summation reward case just works as in [18] and the nonlinear reward case relies on the properties of monotonicity and bounded smoothness used in [8].

## 4 Improved SDCB Algorithm by Discretization

In Section 3, we have shown that our algorithm SDCB achieves near-optimal regret bounds. However, that algorithm might suffer from large running time and memory usage. Note that, in the $t$-th round, an arm $i$ might have been observed $t - 1$ times already, and it is possible that all the observed values from arm $i$ are different (e.g., when arm $i$'s outcome distribution $D_i$ is continuous). In such case, it takes $\Theta(t)$ space to store the empirical CDF $\hat{F}_i$ of the observed outcomes from arm $i$, and both calculating the stochastically dominant CDF $\underline{F}_i$ and updating $\hat{F}_i$ take $\Theta(t)$ time. Therefore, the worst-case space usage of SDCB in $T$ rounds is $\Theta(T)$, and the worst-case running time is $\Theta(T^2)$ (ignoring the dependence on $m$ and $K$; here we do not count the time and space used by the offline computation oracle.

In this section, we propose an improved algorithm Lazy-SDCB which reduces the worst-case memory usage and running time to $O(\sqrt{T})$ and $O(T^{3/2})$, respectively, while preserving the $O(\sqrt{T \log T})$ distribution-independent regret bound. To this end, we need an additional assumption on the reward function:

**Assumption 4** (Lipschitz-continuous reward function)**.** *There exists $C > 0$ such that for any $S \in \mathcal{F}$ and any $x, x' \in [0, 1]^m$, we have $|R(x, S) - R(x', S)| \le C\|x_S - x_S'\|_1$, where $\|x_S - x_S'\|_1 = \sum_{i \in S} |x_i - x_i'|$.*

---

**Algorithm 2** Lazy-SDCB with known time horizon

---
**Input:** time horizon $T$
1: $s \leftarrow \lceil \sqrt{T} \rceil$
2: $I_j \leftarrow \begin{cases} [0, \frac{1}{s}], & j = 1 \\ (\frac{j-1}{s}, \frac{j}{s}], & j = 2, \ldots, s \end{cases}$
3: Invoke SDCB (Algorithm 1) for $T$ rounds, with the following change: whenever observing an outcome $x$ (from any arm), find $j \in [s]$ such that $x \in I_j$, and regard this outcome as $\frac{j}{s}$

---

---

**Algorithm 3** Lazy-SDCB without knowing the time horizon

---
1: $q \leftarrow \lceil \log_2 m \rceil$
2: In rounds $1, 2, \ldots, 2^q$, invoke Algorithm 2 with input $T = 2^q$
3: **for** $k = q, q+1, q+2, \ldots$ **do**
4:     In rounds $2^k + 1, 2^k + 2, \ldots, 2^{k+1}$, invoke Algorithm 2 with input $T = 2^k$
5: **end for**

---

We first describe the algorithm when the time horizon $T$ is known in advance. The algorithm is summarized in Algorithm 2. We perform a *discretization* on the distribution $D = D_1 \times \cdots \times D_m$ to obtain a discrete distribution $\tilde{D} = \tilde{D}_1 \times \cdots \times \tilde{D}_m$ such that (i) for $\tilde{X} \sim \tilde{D}$, $\tilde{X}_1, \ldots, \tilde{X}_m$ are also mutually independent, and (ii) every $\tilde{D}_i$ is supported on a set of equally-spaced values $\{\frac{1}{s}, \frac{2}{s}, \ldots, 1\}$, where $s$ is set to be $\lceil \sqrt{T} \rceil$. Specifically, we partition $[0, 1]$ into $s$ intervals: $I_1 = [0, \frac{1}{s}], I_2 = (\frac{1}{s}, \frac{2}{s}], \ldots, I_{s-1} = (\frac{s-2}{s}, \frac{s-1}{s}], I_s = (\frac{s-1}{s}, 1]$, and define $\tilde{D}_i$ as

$$\Pr_{\tilde{X}_i \sim \tilde{D}_i} [\tilde{X}_i = j/s] = \Pr_{X_i \sim D_i} [X_i \in I_j], \qquad j = 1, \ldots, s.$$

For the CMAB problem $([m], \mathcal{F}, D, R)$, our algorithm "pretends" that the outcomes are drawn from $\tilde{D}$ instead of $D$, by replacing any outcome $x \in I_j$ by $\frac{j}{s}$ ($\forall j \in [s]$), and then applies SDCB to the problem $([m], \mathcal{F}, \tilde{D}, R)$. Since each $\tilde{D}_i$ has a known support $\{\frac{1}{s}, \frac{2}{s}, \ldots, 1\}$, the algorithm only needs to maintain the number of occurrences of each support value in order to obtain the empirical CDF of all the observed outcomes from arm $i$. Therefore, all the operations in a round can be done using $O(s) = O(\sqrt{T})$ time and space, and the total time and space used by Lazy-SDCB are $O(T^{3/2})$ and $O(\sqrt{T})$, respectively.

The discretization parameter $s$ in Algorithm 2 depends on the time horizon $T$, which is why Algorithm 2 has to know $T$ in advance. We can use the doubling trick to avoid the dependency on $T$. We present such an algorithm (without knowing $T$) in Algorithm 3. It is easy to see that Algorithm 3 has the same asymptotic time and space usages as Algorithm 2.

**Regret Bounds.** We show that both Algorithm 2 and Algorithm 3 achieve $O(\sqrt{T \log T})$ distribution-independent regret bounds. The full proofs are given in the supplementary material. Recall that $C$ is the coefficient in the Lipschitz condition in Assumption 4.

**Theorem 2.** *Suppose the time horizon $T$ is known in advance. Then the $\alpha$-approximation regret of Algorithm 2 in $T$ rounds is at most*

$$93M\sqrt{mKT \ln T} + 2CK\sqrt{T} + \left( \frac{\pi^2}{3} + 1 \right) \alpha M m.$$

*Proof Sketch.* The regret consists of two parts: (i) the regret for the discretized CMAB problem $([m], \mathcal{F}, \tilde{D}, R)$, and (ii) the error due to discretization. We directly apply Theorem 1 for the first part. For the second part, a key step is to show $|r_D(S) - r_{\tilde{D}}(S)| \leq CK/s$ for all $S \in \mathcal{F}$ (see the supplementary material). $\square$

**Theorem 3.** *For any time horizon $T \geq 2$, the $\alpha$-approximation regret of Algorithm 3 in $T$ rounds is at most*

$$318M\sqrt{mKT \ln T} + 7CK\sqrt{T} + 10\alpha M m \ln T.$$

# 5 Applications

We describe the $K$-MAX problem and the class of expected utility maximization problems as applications of our general CMAB framework.

**The $K$-MAX Problem.** In this problem, the player is allowed to select at most $K$ arms from the set of $m$ arms in each round, and the reward is the maximum one among the outcomes from the selected arms. In other words, the set of feasible super arms is $\mathcal{F} = \{S \subseteq [m] \mid |S| \leq K\}$, and the reward function is $R(x, S) = \max_{i \in S} x_i$. It is easy to verify that this reward function satisfies Assumptions 2, 3 and 4 with $M = C = 1$.

Now we consider the corresponding offline $K$-MAX problem of selecting at most $K$ arms from $m$ independent arms, with the largest expected reward. It can be implied by a result in [14] that finding the exact optimal solution is NP-hard, so we resort to approximation algorithms. We can show, using submodularity, that a simple greedy algorithm can achieve a $(1 - 1/e)$-approximation. Furthermore, we give the first PTAS for this problem. Our PTAS can be generalized to constraints other than the cardinality constraint $|S| \leq K$, including $s$-$t$ simple paths, matchings, knapsacks, etc. The algorithms and corresponding proofs are given in the supplementary material.

**Theorem 4.** *There exists a PTAS for the offline $K$-MAX problem. In other words, for any constant $\epsilon > 0$, there is a polynomial-time $(1 - \epsilon)$-approximation algorithm for the offline $K$-MAX problem.*

We thus can apply our SDCB algorithm to the $K$-MAX bandit problem and obtain $O(\log T)$ distribution-dependent and $\tilde{O}(\sqrt{T})$ distribution-independent regret bounds according to Theorem 1, or can apply Lazy-SDCB to get $\tilde{O}(\sqrt{T})$ distribution-independent bound according to Theorem 2 or 3.

Streeter and Golovin [26] study an *online submodular maximization* problem in the oblivious adversary model. In particular, their result can cover the stochastic $K$-MAX bandit problem as a special case, and an $O(K\sqrt{mT \log m})$ upper bound on the $(1 - 1/e)$-regret can be shown. While the techniques in [26] can only give a bound on the $(1 - 1/e)$-approximation regret for $K$-MAX, we can obtain the first $\tilde{O}(\sqrt{T})$ bound on the $(1 - \epsilon)$-approximation regret for any constant $\epsilon > 0$, using our PTAS as the offline oracle. Even when we use the simple greedy algorithm as the oracle, our experiments show that SDCB performs significantly better than the algorithm in [26] (see the supplementary material).

**Expected Utility Maximization.** Our framework can also be applied to reward functions of the form $R(x, S) = u(\sum_{i \in S} x_i)$, where $u(\cdot)$ is an increasing utility function. The corresponding offline problem is to maximize the expected utility $\mathbb{E}[u(\sum_{i \in S} x_i)]$ subject to a feasibility constraint $S \in \mathcal{F}$. Note that if $u$ is nonlinear, the expected utility may not be a function of the means of the arms in $S$. Following the celebrated von Neumann-Morgenstern expected utility theorem, nonlinear utility functions have been extensively used to capture risk-averse or risk-prone behaviors in economics (see e.g., [11]), while linear utility functions correspond to risk-neutrality.

Li and Deshpande [20] obtain a PTAS for the expected utility maximization (EUM) problem for several classes of utility functions (including for example increasing concave functions which typically indicate risk-averseness), and a large class of feasibility constraints (including cardinality constraint, $s$-$t$ simple paths, matchings, and knapsacks). Similar results for other utility functions and feasibility constraints can be found in [27, 21, 4]. In the online problem, we can apply our algorithms, using their PTASs as the offline oracle. Again, we can obtain the first tight regret bounds on the $(1 - \epsilon)$-approximation regret for any $\epsilon > 0$, for the class of online EUM problems.

## Acknowledgments

Wei Chen was supported in part by the National Natural Science Foundation of China (Grant No. 61433014). Jian Li and Yu Liu were supported in part by the National Basic Research Program of China grants 2015CB358700, 2011CBA00300, 2011CBA00301, and the National NSFC grants 61033001, 61361136003. The authors would like to thank Tor Lattimore for referring to us the DKW inequality.

## Footnotes

[1]We understand that the first-price auction is not truthful, but this example is only for illustrative purpose for the $\max()$ function.

[2]The Lipschitz-continuity assumption is only made for Lazy-SDCB. See Section 4.

[3]$[0, 1]^S$ is isomorphic to $[0, 1]^{|S|}$; the coordinates in $[0, 1]^S$ are indexed by elements in $S$.

[4]Note that $S_t^{\mathcal{A}}$ may be random due to the random outcomes in previous rounds and the possible randomness used by $\mathcal{A}$.

[5]Without loss of generality, we assume that each arm $i \in [m]$ is contained in at least one super arm.

[6]We remark that while $\underline{F}_i(x)$ is a numerical lower confidence bound on $F_i(x)$ for all $x \in [0, 1]$, at the distribution level, $\underline{D}_i$ serves as a "stochastically dominant (upper) confidence bound" on $D_i$.

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
