[Supplementary Material · nips2016_paper903_supplementary.pdf]

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

[7]We use $\mathbb{1}\{\cdot\}$ to denote the indicator function, i.e., $\mathbb{1}\{\mathcal{H}\} = 1$ if an event $\mathcal{H}$ happens, and $\mathbb{1}\{\mathcal{H}\} = 0$ if it does not happen.

[8] It is not hard to see the signature of $\max_{k \in [h]} B_k$ is exactly $\mathsf{sg}$.

[9] In the exact version of a problem, we ask for a feasible set $S$ such that total weight of $S$ is exactly a given target value $B$. For example, in the exact spanning tree problem where each edge has an integer weight, we would like to find a spanning tree of weight exactly $B$.

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

# Appendix

## A   Missing Proofs from Section 3

### A.1   Proof of Theorem 1

We present the proof of Theorem 1 in four steps. In Section A.1.1, we review the $L_1$ distance between two distributions and present a property of it. In Section A.1.2, we review the Dvoretzky-Kiefer-Wolfowitz (DKW) inequality, which is a strong concentration result for empirical CDFs. In Section A.1.3, we prove some key technical lemmas. Then we complete the proof of Theorem 1 in Section A.1.4.

### A.1.1   The $L_1$ Distance between Two Probability Distributions

For simplicity, we only consider discrete distributions with finite supports – this will be enough for our purpose.

Let $P$ be a probability distribution. For any $x$, let $P(x) = \Pr_{X \sim P}[X = x]$. We write $P = P_1 \times P_2 \times \cdots \times P_n$ if the (multivariate) random variable $X \sim P$ can be written as $X = (X_1, X_2, \ldots, X_n)$, where $X_1, \ldots, X_n$ are mutually independent and $X_i \sim P_i$ ($\forall i \in [n]$).

For two distributions $P$ and $Q$, their $L_1$ *distance* is defined as

$$L_1(P, Q) = \sum_x |P(x) - Q(x)|,$$

where the summation is taken over $x \in \mathsf{supp}(P) \cup \mathsf{supp}(Q)$.

The $L_1$ distance has the following property. It is a folklore result and we provide a proof for completeness.

**Lemma 1.** *Let $P = P_1 \times P_2 \times \cdots \times P_n$ and $Q = Q_1 \times Q_2 \times \cdots \times Q_n$ be two probability distributions. Then we have*

$$L_1(P, Q) \leq \sum_{i=1}^{n} L_1(P_i, Q_i). \tag{1}$$

*Proof.* We prove (1) by induction on $n$.

When $n = 2$, we have

$$
\begin{aligned}
L_1(P, Q) &= \sum_x \sum_y |P(x, y) - Q(x, y)| \\
&= \sum_x \sum_y |P_1(x)P_2(y) - Q_1(x)Q_2(y)| \\
&\leq \sum_x \sum_y (|P_1(x)P_2(y) - P_1(x)Q_2(y)| + |P_1(x)Q_2(y) - Q_1(x)Q_2(y)|) \\
&= \sum_x P_1(x) \sum_y |P_2(y) - Q_2(y)| + \sum_y Q_2(y) \sum_x |P_1(x) - Q_1(x)| \\
&= 1 \cdot L_1(P_2, Q_2) + 1 \cdot L_1(P_1, Q_1) \\
&= \sum_{i=1}^{2} L_1(P_i, Q_i).
\end{aligned}
$$

Here the summation is taken over $x \in \mathsf{supp}(P_1) \cup \mathsf{supp}(Q_1)$ and $y \in \mathsf{supp}(P_2) \cup \mathsf{supp}(Q_2)$.

Suppose (1) is proved for $n = k - 1$ ($k \geq 3$). When $n = k$, using the results for $n = k - 1$ and $n = 2$, we get

$$L_1(P, Q) \leq \sum_{i=1}^{k-2} L_1(P_i, Q_i) + L_1(P_{k-1} \times P_k, Q_{k-1} \times Q_k)$$

$$\leq \sum_{i=1}^{k-2} L_1(P_i, Q_i) + L_1(P_{k-1}, Q_{k-1}) + L_1(P_k, Q_k)$$

$$= \sum_{i=1}^{k} L_1(P_i, Q_i).$$

This completes the proof. $\qquad\qquad\qquad\qquad\qquad\qquad\qquad\qquad\qquad\qquad$ □

### A.1.2 The DKW Inequality

Consider a distribution $D$ with CDF $F(x)$. Let $\hat{F}_n(x)$ be the empirical CDF of $n$ i.i.d. samples $X_1, \ldots, X_n$ drawn from $D$, i.e., $\hat{F}_n(x) = \frac{1}{n}\sum_{i=1}^{n} \mathbb{1}\{X_i \leq x\}$ $(x \in \mathbb{R})$.[7] Then we have:

**Lemma 2** (Dvoretzky-Kiefer-Wolfowitz inequality [10, 24])**.** *For any $\epsilon > 0$ and any $n \in \mathbb{Z}_+$, we have*

$$\Pr\left[\sup_{x \in \mathbb{R}} \left|\hat{F}_n(x) - F(x)\right| \geq \epsilon\right] \leq 2e^{-2n\epsilon^2}.$$

Note that for any fixed $x \in \mathbb{R}$, from the Chernoff bound we have $\Pr\left[\left|\hat{F}_n(x) - F(x)\right| \geq \epsilon\right] \leq 2e^{-2n\epsilon^2}$. The DKW inequality states a stronger guarantee that the Chernoff concentration holds simultaneously for all $x \in \mathbb{R}$.

### A.1.3 Technical Lemmas

The following lemma describes some properties of the expected reward $r_P(S) = \mathbb{E}_{X \sim P}[R(X, S)]$.

**Lemma 3.** *Let $P = P_1 \times \cdots \times P_m$ and $P' = P_1' \times \cdots \times P_m'$ be two probability distributions over $[0,1]^m$. Let $F_i$ and $F_i'$ be the CDFs of $P_i$ and $P_i'$, respectively $(i = 1, \ldots, m)$. Suppose each $P_i$ $(i \in [m])$ is a discrete distribution with finite support.*

*(i) If for any $i \in [m], x \in [0,1]$ we have $F_i'(x) \leq F_i(x)$, then for any super arm $S \in \mathcal{F}$, we have*

$$r_{P'}(S) \geq r_P(S).$$

*(ii) If for any $i \in [m], x \in [0,1]$ we have $F_i(x) - F_i'(x) \leq \Lambda_i$ $(\Lambda_i > 0)$, then for any super arm $S \in \mathcal{F}$, we have*

$$r_{P'}(S) - r_P(S) \leq 2M \sum_{i \in S} \Lambda_i.$$

*Proof.* It is easy to see why (i) is true. If we have $F_i'(x) \leq F_i(x)$ for all $i \in [m]$ and $x \in [0,1]$, then for all $i$, $P_i'$ has first-order stochastic dominance over $P_i$. When we change the distribution from $P_i$ into $P_i'$, we are moving some probability mass from smaller values to larger values. Recall that the reward function $R(x, S)$ has a monotonicity property (Assumption 3): if $x$ and $x'$ are two vectors in $[0,1]^m$ such that $x_i \leq x_i'$ for all $i \in [m]$, then $R(x, S) \leq R(x', S)$ for all $S \in \mathcal{F}$. Therefore we have $r_P(S) \leq r_{P'}(S)$ for all $S \in \mathcal{F}$.

Now we prove (ii). Without loss of generality, we assume $S = \{1, 2, \ldots, n\}$ $(n \leq m)$. Let $P'' = P_1'' \times \cdots \times P_m''$ be a distribution over $[0,1]^m$ such that the CDF of $P_i''$ is the following:

$$F_i''(x) = \begin{cases} \max\{F_i(x) - \Lambda_i, 0\}, & 0 \leq x < 1, \\ 1, & x = 1. \end{cases} \tag{2}$$

It is easy to see that $F_i''(x) \leq F_i'(x)$ for all $i \in [m]$ and $x \in [0,1]$. Thus from the result in (i) we have

$$r_{P'}(S) \leq r_{P''}(S). \tag{3}$$

Let $\mathrm{supp}(P_i) = \{v_{i,1}, v_{i,2}, \ldots, v_{i,s_i}\}$ where $0 \leq v_{i,1} < \cdots < v_{i,s_i} \leq 1$. Define $P_S = P_1 \times P_2 \times \cdots \times P_n$, and define $P_S'$ and $P_S''$ similarly. Recall that the reward function $R(x, S)$ can be written as $R_S(x_S) = R_S(x_1, \ldots, x_n)$. Then we have

$$
\begin{aligned}
&r_{P''}(S) - r_P(S) \\
&= \sum_{x_1,\ldots,x_n} R_S(x_1, \ldots, x_n) P_S''(x_1, \ldots, x_n) - \sum_{x_1,\ldots,x_n} R_S(x_1, \ldots, x_n) P_S(x_1, \ldots, x_n) \\
&= \sum_{x_1,\ldots,x_n} R_S(x_1, \ldots, x_n) \cdot (P_S''(x_1, \ldots, x_n) - P_S(x_1, \ldots, x_n)) \\
&\leq \sum_{x_1,\ldots,x_n} M \cdot |P_S''(x_1, \ldots, x_n) - P_S(x_1, \ldots, x_n)| \\
&= M \cdot L_1(P_S'', P_S),
\end{aligned}
$$

where the summation is taken over $x_i \in \{v_{i,1}, \ldots, v_{i,s_i}\}$ ($\forall i \in S$). Then using Lemma 1 we obtain

$$
r_{P''}(S) - r_P(S) \leq M \cdot \sum_{i \in S} L_1(P_i'', P_i). \tag{4}
$$

Now we give an upper bound on $L_1(P_i'', P_i)$ for each $i$. Let $F_{i,j} = F_i(v_{i,j})$, $F_{i,j}'' = F_i''(v_{i,j})$, and $F_{i,0} = F_{i,0}'' = 0$. We have

$$
\begin{aligned}
L_1(P_i'', P_i) &= \sum_{j=1}^{s_i} |P_i''(v_{i,j}) - P_i(v_{i,j})| \\
&= \sum_{j=1}^{s_i} \left| (F_{i,j}'' - F_{i,j-1}'') - (F_{i,j} - F_{i,j-1}) \right| \\
&= \sum_{j=1}^{s_i} \left| (F_{i,j} - F_{i,j}'') - (F_{i,j-1} - F_{i,j-1}'') \right|.
\end{aligned} \tag{5}
$$

In fact, for all $1 \leq j < s_i$, we have $F_{i,j} - F_{i,j}'' \geq F_{i,j-1} - F_{i,j-1}''$. To see this, consider two cases:

- If $F_{i,j} < \Lambda_i$, then we have $F_{i,j-1} \leq F_{i,j} < \Lambda_i$. By definition (2) we have $F_{i,j}'' = F_{i,j-1}'' = 0$. Thus $F_{i,j} - F_{i,j}'' = F_{i,j} \geq F_{i,j-1} = F_{i,j-1} - F_{i,j-1}''$.

- If $F_{i,j} \geq \Lambda_i$, then by definition (2) we have $F_{i,j} - F_{i,j}'' = \Lambda_i \geq F_{i,j-1} - F_{i,j-1}''$.

Therefore (5) becomes

$$
\begin{aligned}
L_1(P_i'', P_i) &= \sum_{j=1}^{s_i-1} \left( (F_{i,j} - F_{i,j}'') - (F_{i,j-1} - F_{i,j-1}'') \right) + \left| (1 - 1) - (F_{i,s_i-1} - F_{i,s_i-1}'') \right| \\
&= F_{i,s_i-1} - F_{i,s_i-1}'' + \left| F_{i,s_i-1} - F_{i,s_i-1}'' \right| \\
&= 2 \left( F_{i,s_i-1} - F_{i,s_i-1}'' \right) \\
&\leq 2\Lambda_i,
\end{aligned} \tag{6}
$$

where the last inequality is due to (2).

We complete the proof of the lemma by combining (3), (4) and (6):

$$
r_{P'}(S) - r_P(S) \leq r_{P''}(S) - r_P(S) \leq M \cdot \sum_{i \in S} L_1(P_i'', P_i) \leq 2M \sum_{i \in S} \Lambda_i. \qquad \square
$$

The following lemma is similar to Lemma 1 in [18]. We will use some additional notation:

- For $t \geq m + 1$ and $i \in [m]$, let $T_{i,t}$ be the value of counter $T_i$ right after the $t$-th round of SDCB. In other words, $T_{i,t}$ is the number of observed outcomes from arm $i$ in the first $t$ rounds.

- Let $S_t$ be the super arm selected by SDCB in the $t$-th round.

**Lemma 4.** *Define an event in each round $t$ $(m + 1 \leq t \leq T)$:*

$$\mathcal{H}_t = \left\{ 0 < \Delta_{S_t} \leq 4M \cdot \sum_{i \in S_t} \sqrt{\frac{3 \ln t}{2 T_{i,t-1}}} \right\}. \tag{7}$$

*Then the $\alpha$-approximation regret of SDCB in $T$ rounds is at most*

$$\mathbb{E} \left[ \sum_{t=m+1}^{T} \mathbb{1}\{\mathcal{H}_t\} \Delta_{S_t} \right] + \left( \frac{\pi^2}{3} + 1 \right) \alpha M m.$$

*Proof.* Let $F_i$ be the CDF of $D_i$. Let $\hat{F}_{i,l}$ be the empirical CDF of the first $l$ observations from arm $i$. For $m + 1 \leq t \leq T$, define an event

$$\mathcal{E}_t = \left\{ \text{there exists } i \in [m] \text{ such that } \sup_{x \in [0,1]} \left| \hat{F}_{i,T_{i,t-1}}(x) - F_i(x) \right| \geq \sqrt{\frac{3 \ln t}{2 T_{i,t-1}}} \right\},$$

which means that the empirical CDF $\hat{F}_i$ is not close enough to the true CDF $F_i$ at the *beginning* of the $t$-th round.

Recall that we have $S^* = \operatorname{argmax}_{S \in \mathcal{F}} \{ r_D(S) \}$ and $\Delta_S = \max\{ \alpha \cdot r_D(S^*) - r_D(S), 0 \}$ $(S \in \mathcal{F})$. We bound the $\alpha$-approximation regret of SDCB as

$$\begin{aligned}
\mathsf{Reg}_{D,\alpha}^{\mathsf{SDCB}}(T) &= \sum_{t=1}^{T} \mathbb{E} \left[ \alpha \cdot r_D(S^*) - r_D(S_t) \right] \leq \sum_{t=1}^{T} \mathbb{E} \left[ \Delta_{S_t} \right] \\
&= \mathbb{E} \left[ \sum_{t=1}^{m} \Delta_{S_t} \right] + \mathbb{E} \left[ \sum_{t=m+1}^{T} \mathbb{1}\{\mathcal{E}_t\} \Delta_{S_t} \right] + \mathbb{E} \left[ \sum_{t=m+1}^{T} \mathbb{1}\{\neg\mathcal{E}_t\} \Delta_{S_t} \right],
\end{aligned} \tag{8}$$

where $\neg\mathcal{E}_t$ is the complement of event $\mathcal{E}_t$.

We separately bound each term in (8).

(a) the first term

The first term in (8) can be trivially bounded as

$$\mathbb{E} \left[ \sum_{t=1}^{m} \Delta_{S_t} \right] \leq \sum_{t=1}^{m} \alpha \cdot r_D(S^*) \leq m \cdot \alpha M. \tag{9}$$

(b) the second term

By the DKW inequality we know that for any $i \in [m], l \geq 1, t \geq m + 1$ we have

$$\Pr \left[ \sup_{x \in [0,1]} \left| \hat{F}_{i,l}(x) - F_i(x) \right| \geq \sqrt{\frac{3 \ln t}{2l}} \right] \leq 2 e^{-2l \cdot \frac{3 \ln t}{2l}} = 2 e^{-3 \ln t} = 2 t^{-3}.$$

Therefore

$$\begin{aligned}
\mathbb{E} \left[ \sum_{t=m+1}^{T} \mathbb{1}\{\mathcal{E}_t\} \right] &\leq \sum_{t=m+1}^{T} \sum_{i=1}^{m} \sum_{l=1}^{t-1} \Pr \left[ \left| \hat{F}_{i,j,l} - F_{i,j} \right| \geq \sqrt{\frac{3 \ln t}{2l}} \right] \\
&\leq \sum_{t=m+1}^{T} \sum_{i=1}^{m} \sum_{l=1}^{t-1} 2 t^{-3} \\
&\leq 2m \sum_{t=m+1}^{T} t^{-2} \\
&\leq \frac{\pi^2}{3} m,
\end{aligned}$$

and then the second term in (8) can be bounded as

$$\mathbb{E}\left[\sum_{t=m+1}^{T}\mathbb{1}\{\mathcal{E}_t\}\Delta_{S_t}\right] \leq \frac{\pi^2}{3}m \cdot (\alpha \cdot r_D(S^*)) \leq \frac{\pi^2}{3}\alpha Mm. \tag{10}$$

(c) the third term

We fix $t > m$ and first assume $\neg\mathcal{E}_t$ happens. Let $c_i = \sqrt{\frac{3\ln t}{2T_{i,t-1}}}$ for each $i \in [m]$. Since $\neg\mathcal{E}_t$ happens, we have

$$\left|\hat{F}_{i,T_{i,t-1}}(x) - F_i(x)\right| < c_i \qquad \forall i \in [m], x \in [0,1]. \tag{11}$$

Recall that in round $t$ of SDCB (Algorithm 1), the input to the oracle is $\underline{D} = \underline{D}_1 \times \cdots \times \underline{D}_m$, where the CDF $\underline{F}_i$ of $\underline{D}_i$ is

$$\underline{F}_i(x) = \begin{cases} \max\{\hat{F}_{i,T_{i,t-1}}(x) - c_i, 0\}, & 0 \leq x < 1, \\ 1, & x = 1. \end{cases} \tag{12}$$

From (11) and (12) we know that $\underline{F}_i(x) \leq F_i(x) \leq \underline{F}_i(x) + 2c_i$ for all $i \in [m], x \in [0,1]$. Thus, from Lemma 3 (i) we have

$$r_D(S) \leq r_{\underline{D}}(S) \qquad \forall S \in \mathcal{F}, \tag{13}$$

and from Lemma 3 (ii) we have

$$r_{\underline{D}}(S) \leq r_D(S) + 2M\sum_{i \in S}2c_i \qquad \forall S \in \mathcal{F}. \tag{14}$$

Also, from the fact that the algorithm chooses $S_t$ in the $t$-th round, we have

$$r_{\underline{D}}(S_t) \geq \alpha \cdot \max_{S \in \mathcal{F}}\{r_{\underline{D}}(S)\} \geq \alpha \cdot r_{\underline{D}}(S^*). \tag{15}$$

From (13), (14) and (15) we have

$$\alpha \cdot r_D(S^*) \leq \alpha \cdot r_{\underline{D}}(S^*) \leq r_{\underline{D}}(S_t) \leq r_D(S_t) + 2M\sum_{i \in S_t}2c_i,$$

which implies

$$\Delta_{S_t} \leq 4M\sum_{i \in S_t}c_i.$$

Therefore, when $\neg\mathcal{E}_t$ happens, we always have $\Delta_{S_t} \leq 4M\sum_{i \in S_t}c_i$. In other words,

$$\neg\mathcal{E}_t \implies \left\{\Delta_{S_t} \leq 4M\sum_{i \in S_t}\sqrt{\frac{3\ln t}{2T_{i,t-1}}}\right\}.$$

This implies

$$\{\neg\mathcal{E}_t, \Delta_{S_t} > 0\} \implies \left\{0 < \Delta_{S_t} \leq 4M\sum_{i \in S_t}\sqrt{\frac{3\ln t}{2T_{i,t-1}}}\right\} = \mathcal{H}_t.$$

Hence, the third term in (8) can be bounded as

$$\mathbb{E}\left[\sum_{t=m+1}^{T}\mathbb{1}\{\neg\mathcal{E}_t\}\Delta_{S_t}\right] = \mathbb{E}\left[\sum_{t=m+1}^{T}\mathbb{1}\{\neg\mathcal{E}_t, \Delta_{S_t} > 0\}\Delta_{S_t}\right] \leq \mathbb{E}\left[\sum_{t=m+1}^{T}\mathbb{1}\{\mathcal{H}_t\}\Delta_{S_t}\right]. \tag{16}$$

Finally, by combining (8), (9), (10) and (16) we have

$$\mathsf{Reg}_{D,\alpha}^{\mathsf{SDCB}}(T) \leq \mathbb{E}\left[\sum_{t=m+1}^{T}\mathbb{1}\{\mathcal{H}_t\}\Delta_{S_t}\right] + \left(\frac{\pi^2}{3}+1\right)\alpha Mm,$$

completing the proof of the lemma. $\qquad\square$

### A.1.4 Finishing the Proof of Theorem 1

Lemma 4 is very similar to Lemma 1 in [18]. We now apply the counting argument in [18] to finish the proof of Theorem 1.

From Lemma 4 we know that it remains to bound $\mathbb{E}\left[\sum_{t=m+1}^{T} \mathbb{1}\{\mathcal{H}_t\}\Delta_{S_t}\right]$, where $\mathcal{H}_t$ is defined in (7).

Define two decreasing sequences of positive constants

$$1 = \beta_0 > \beta_1 > \beta_2 > \ldots$$
$$\alpha_1 > \alpha_2 > \ldots$$

such that $\lim_{k\to\infty} \alpha_k = \lim_{k\to\infty} \beta_k = 0$. We choose $\{\alpha_k\}$ and $\{\beta_k\}$ as in Theorem 4 of [18], which satisfy

$$\sqrt{6}\sum_{k=1}^{\infty} \frac{\beta_{k-1} - \beta_k}{\sqrt{\alpha_k}} \leq 1 \tag{17}$$

and

$$\sum_{k=1}^{\infty} \frac{\alpha_k}{\beta_k} < 267. \tag{18}$$

For $t \in \{m+1, \ldots, T\}$ and $k \in \mathbb{Z}_+$, let

$$m_{k,t} = \begin{cases} \alpha_k \left(\frac{2MK}{\Delta_{S_t}}\right)^2 \ln T & \Delta_{S_t} > 0, \\ +\infty & \Delta_{S_t} = 0, \end{cases}$$

and

$$A_{k,t} = \{i \in S_t \mid T_{i,t-1} \leq m_{k,t}\}.$$

Then we define an event

$$\mathcal{G}_{k,t} = \{|A_{k,t}| \geq \beta_k K\},$$

which means "in the $t$-th round, at least $\beta_k K$ arms in $S_t$ had been observed at most $m_{k,t}$ times."

**Lemma 5.** *In the $t$-th round ($m+1 \leq t \leq T$), if event $\mathcal{H}_t$ happens, then there exists $k \in \mathbb{Z}_+$ such that event $\mathcal{G}_{k,t}$ happens.*

*Proof.* Assume that $\mathcal{H}_t$ happens and that none of $\mathcal{G}_{1,t}, \mathcal{G}_{2,t}, \ldots$ happens. Then $|A_{k,t}| < \beta_k K$ for all $k \in \mathbb{Z}_+$.

Let $A_{0,t} = S_t$ and $\bar{A}_{k,t} = S_t \setminus A_{k,t}$ for $k \in \mathbb{Z}_+ \cup \{0\}$. It is easy to see $\bar{A}_{k-1,t} \subseteq \bar{A}_{k,t}$ for all $k \in \mathbb{Z}_+$. Note that $\lim_{k\to\infty} m_{k,t} = 0$. Thus there exists $N \in \mathbb{Z}_+$ such that $\bar{A}_{k,t} = S_t$ for all $k \geq N$, and then we have $S_t = \bigcup_{k=1}^{\infty} (\bar{A}_{k,t} \setminus \bar{A}_{k-1,t})$. Finally, note that for all $i \in \bar{A}_{k,t}$, we have $T_{i,t-1} > m_{k,t}$. Therefore

$$\sum_{i \in S_t} \frac{1}{\sqrt{T_{i,t-1}}} = \sum_{k=1}^{\infty} \sum_{i \in \bar{A}_{k,t} \setminus \bar{A}_{k-1,t}} \frac{1}{\sqrt{T_{i,t-1}}} \leq \sum_{k=1}^{\infty} \sum_{i \in \bar{A}_{k,t} \setminus \bar{A}_{k-1,t}} \frac{1}{\sqrt{m_{k,t}}}$$

$$= \sum_{k=1}^{\infty} \frac{|\bar{A}_{k,t} \setminus \bar{A}_{k-1,t}|}{\sqrt{m_{k,t}}} = \sum_{k=1}^{\infty} \frac{|A_{k-1,t} \setminus A_{k,t}|}{\sqrt{m_{k,t}}} = \sum_{k=1}^{\infty} \frac{|A_{k-1,t}| - |A_{k,t}|}{\sqrt{m_{k,t}}}$$

$$= \frac{|S_t|}{\sqrt{m_{1,t}}} + \sum_{k=1}^{\infty} |A_{k,t}| \left(\frac{1}{\sqrt{m_{k+1,t}}} - \frac{1}{\sqrt{m_{k,t}}}\right)$$

$$< \frac{K}{\sqrt{m_{1,t}}} + \sum_{k=1}^{\infty} \beta_k K \left(\frac{1}{\sqrt{m_{k+1,t}}} - \frac{1}{\sqrt{m_{k,t}}}\right)$$

$$= \sum_{k=1}^{\infty} \frac{(\beta_{k-1} - \beta_k)K}{\sqrt{m_{k,t}}}.$$

Note that we assume $\mathcal{H}_t$ happens. Then we have

$$\Delta_{S_t} \leq 4M \cdot \sum_{i \in S_t} \sqrt{\frac{3 \ln t}{2 T_{i,t-1}}} \leq 2M \sqrt{6 \ln T} \cdot \sum_{i \in S_t} \frac{1}{\sqrt{T_{i,t-1}}}$$

$$< 2M \sqrt{6 \ln T} \cdot \sum_{k=1}^{\infty} \frac{(\beta_{k-1} - \beta_k) K}{\sqrt{m_{k,t}}} = \sqrt{6} \sum_{k=1}^{\infty} \frac{\beta_{k-1} - \beta_k}{\sqrt{\alpha_k}} \cdot \Delta_{S_t} \leq \Delta_{S_t},$$

where the last inequality is due to (17). We reach a contradiction here. The proof of the lemma is completed. $\square$

By Lemma 5 we have

$$\sum_{t=m+1}^{T} \mathbb{1}\{\mathcal{H}_t\} \Delta_{S_t} \leq \sum_{k=1}^{\infty} \sum_{t=m+1}^{T} \mathbb{1}\{\mathcal{G}_{k,t}, \Delta_{S_t} > 0\} \Delta_{S_t}.$$

For $i \in [m], k \in \mathbb{Z}_+, t \in \{m+1, \ldots, T\}$, define an event

$$\mathcal{G}_{i,k,t} = \mathcal{G}_{k,t} \wedge \{i \in S_t, T_{i,t-1} \leq m_{k,t}\}.$$

Then by the definitions of $\mathcal{G}_{k,t}$ and $\mathcal{G}_{i,k,t}$ we have

$$\mathbb{1}\{\mathcal{G}_{k,t}, \Delta_{S_t} > 0\} \leq \frac{1}{\beta_k K} \sum_{i \in E_{\mathrm{B}}} \mathbb{1}\{\mathcal{G}_{i,k,t}, \Delta_{S_t} > 0\}.$$

Therefore

$$\sum_{t=m+1}^{T} \mathbb{1}\{\mathcal{H}_t\} \Delta_{S_t} \leq \sum_{i \in E_{\mathrm{B}}} \sum_{k=1}^{\infty} \sum_{t=m+1}^{T} \mathbb{1}\{\mathcal{G}_{i,k,t}, \Delta_{S_t} > 0\} \frac{\Delta_{S_t}}{\beta_k K}.$$

For each arm $i \in E_{\mathrm{B}}$, suppose $i$ is contained in $N_i$ *bad* super arms $S_{i,1}^{\mathrm{B}}, S_{i,2}^{\mathrm{B}}, \ldots, S_{i,N_i}^{\mathrm{B}}$. Let $\Delta_{i,l} = \Delta_{S_{i,l}^{\mathrm{B}}}$ ($l \in [N_i]$). Without loss of generality, we assume $\Delta_{i,1} \geq \Delta_{i,2} \geq \ldots \geq \Delta_{i,N_i}$. Note that $\Delta_{i,N_i} = \Delta_{i,\min}$. For convenience, we also define $\Delta_{i,0} = +\infty$, i.e., $\alpha_k \left( \frac{2MK}{\Delta_{i,0}} \right)^2 = 0$. Then we have

$$\sum_{t=m+1}^{T} \mathbb{1}\{\mathcal{H}_t\} \Delta_{S_t}$$

$$\leq \sum_{i \in E_{\mathrm{B}}} \sum_{k=1}^{\infty} \sum_{t=m+1}^{T} \sum_{l=1}^{N_i} \mathbb{1}\{\mathcal{G}_{i,k,t}, S_t = S_{i,l}^{\mathrm{B}}\} \frac{\Delta_{S_t}}{\beta_k K}$$

$$\leq \sum_{i \in E_{\mathrm{B}}} \sum_{k=1}^{\infty} \sum_{t=m+1}^{T} \sum_{l=1}^{N_i} \mathbb{1}\{T_{i,t-1} \leq m_{k,t}, S_t = S_{i,l}^{\mathrm{B}}\} \frac{\Delta_{i,l}}{\beta_k K}$$

$$= \sum_{i \in E_{\mathrm{B}}} \sum_{k=1}^{\infty} \sum_{t=m+1}^{T} \sum_{l=1}^{N_i} \mathbb{1}\left\{ T_{i,t-1} \leq \alpha_k \left( \frac{2MK}{\Delta_{i,l}} \right)^2 \ln T, S_t = S_{i,l}^{\mathrm{B}} \right\} \frac{\Delta_{i,l}}{\beta_k K}$$

$$= \sum_{i \in E_{\mathrm{B}}} \sum_{k=1}^{\infty} \sum_{t=m+1}^{T} \sum_{l=1}^{N_i} \sum_{j=1}^{l} \mathbb{1}\left\{ \alpha_k \left( \frac{2MK}{\Delta_{i,j-1}} \right)^2 \ln T < T_{i,t-1} \leq \alpha_k \left( \frac{2MK}{\Delta_{i,j}} \right)^2 \ln T, S_t = S_{i,l}^{\mathrm{B}} \right\} \frac{\Delta_{i,l}}{\beta_k K}$$

$$\leq \sum_{i \in E_{\mathrm{B}}} \sum_{k=1}^{\infty} \sum_{t=m+1}^{T} \sum_{l=1}^{N_i} \sum_{j=1}^{l} \mathbb{1}\left\{ \alpha_k \left( \frac{2MK}{\Delta_{i,j-1}} \right)^2 \ln T < T_{i,t-1} \leq \alpha_k \left( \frac{2MK}{\Delta_{i,j}} \right)^2 \ln T, S_t = S_{i,l}^{\mathrm{B}} \right\} \frac{\Delta_{i,j}}{\beta_k K}$$

$$\leq \sum_{i \in E_{\mathrm{B}}} \sum_{k=1}^{\infty} \sum_{t=m+1}^{T} \sum_{l=1}^{N_i} \sum_{j=1}^{N_i} \mathbb{1}\left\{ \alpha_k \left( \frac{2MK}{\Delta_{i,j-1}} \right)^2 \ln T < T_{i,t-1} \leq \alpha_k \left( \frac{2MK}{\Delta_{i,j}} \right)^2 \ln T, S_t = S_{i,l}^{\mathrm{B}} \right\} \frac{\Delta_{i,j}}{\beta_k K}$$

$$\leq \sum_{i \in E_{\mathrm{B}}} \sum_{k=1}^{\infty} \sum_{t=m+1}^{T} \sum_{j=1}^{N_i} \mathbb{1}\left\{ \alpha_k \left( \frac{2MK}{\Delta_{i,j-1}} \right)^2 \ln T < T_{i,t-1} \leq \alpha_k \left( \frac{2MK}{\Delta_{i,j}} \right)^2 \ln T \right\} \frac{\Delta_{i,j}}{\beta_k K}$$

$$\leq \sum_{i \in E_\mathrm{B}} \sum_{k=1}^{\infty} \sum_{j=1}^{N_i} \left( \alpha_k \left( \frac{2MK}{\Delta_{i,j}} \right)^2 \ln T - \alpha_k \left( \frac{2MK}{\Delta_{i,j-1}} \right)^2 \ln T \right) \frac{\Delta_{i,j}}{\beta_k K}$$

$$= 4M^2 K \left( \sum_{k=1}^{\infty} \frac{\alpha_k}{\beta_k} \right) \ln T \cdot \sum_{i \in E_\mathrm{B}} \sum_{j=1}^{N_i} \left( \frac{1}{\Delta_{i,j}^2} - \frac{1}{\Delta_{i,j-1}^2} \right) \Delta_{i,j}$$

$$\leq 1068 M^2 K \ln T \cdot \sum_{i \in E_\mathrm{B}} \sum_{j=1}^{N_i} \left( \frac{1}{\Delta_{i,j}^2} - \frac{1}{\Delta_{i,j-1}^2} \right) \Delta_{i,j},$$

where the last inequality is due to (18).

Finally, for each $i \in E_\mathrm{B}$ we have

$$\sum_{j=1}^{N_i} \left( \frac{1}{\Delta_{i,j}^2} - \frac{1}{\Delta_{i,j-1}^2} \right) \Delta_{i,j} = \frac{1}{\Delta_{i,N_i}} + \sum_{j=1}^{N_i-1} \frac{1}{\Delta_{i,j}^2} (\Delta_{i,j} - \Delta_{i,j+1})$$

$$\leq \frac{1}{\Delta_{i,N_i}} + \int_{\Delta_{i,N_i}}^{\Delta_{i,1}} \frac{1}{x^2} \, \mathrm{d}x$$

$$= \frac{2}{\Delta_{i,N_i}} - \frac{1}{\Delta_{i,1}}$$

$$< \frac{2}{\Delta_{i,\min}}.$$

It follows that

$$\sum_{t=m+1}^{T} \mathbb{1}\{\mathcal{H}_t\} \Delta_{S_t} \leq 1068 M^2 K \ln T \cdot \sum_{i \in E_\mathrm{B}} \frac{2}{\Delta_{i,\min}} = M^2 K \sum_{i \in E_\mathrm{B}} \frac{2136}{\Delta_{i,\min}} \ln T. \quad (19)$$

Combining (19) with Lemma 4, the distribution-dependent regret bound in Theorem 1 is proved.

To prove the distribution-independent bound, we decompose $\sum_{t=m+1}^{T} \mathbb{1}\{\mathcal{H}_t\} \Delta_{S_t}$ into two parts:

$$\sum_{t=m+1}^{T} \mathbb{1}\{\mathcal{H}_t\} \Delta_{S_t} = \sum_{t=m+1}^{T} \mathbb{1}\{\mathcal{H}_t, \Delta_{S_t} \leq \epsilon\} \Delta_{S_t} + \sum_{t=m+1}^{T} \mathbb{1}\{\mathcal{H}_t, \Delta_{S_t} > \epsilon\} \Delta_{S_t}$$

$$\leq \epsilon T + \sum_{t=m+1}^{T} \mathbb{1}\{\mathcal{H}_t, \Delta_{S_t} > \epsilon\} \Delta_{S_t}, \quad (20)$$

where $\epsilon > 0$ is a constant to be determined. The second term can be bounded in the same way as in the proof of the distribution-dependent regret bound, except that we only consider the case $\Delta_{S_t} > \epsilon$. Thus we can replace (19) by

$$\sum_{t=m+1}^{T} \mathbb{1}\{\mathcal{H}_t, \Delta_{S_t} > \epsilon\} \Delta_{S_t} \leq M^2 K \sum_{i \in E_\mathrm{B}, \Delta_{i,\min} > \epsilon} \frac{2136}{\Delta_{i,\min}} \ln T \leq M^2 K m \frac{2136}{\epsilon} \ln T. \quad (21)$$

It follows that

$$\sum_{t=m+1}^{T} \mathbb{1}\{\mathcal{H}_t\} \Delta_{S_t} \leq \epsilon T + M^2 K m \frac{2136}{\epsilon} \ln T.$$

Finally, letting $\epsilon = \sqrt{\frac{2136 M^2 K m \ln T}{T}}$, we get

$$\sum_{t=m+1}^{T} \mathbb{1}\{\mathcal{H}_t\} \Delta_{S_t} \leq 2 \sqrt{2136 M^2 K m T \ln T} < 93 M \sqrt{m K T \ln T}.$$

Combining this with Lemma 4, we conclude the proof of the distribution-independent regret bound in Theorem 1. $\qquad \square$

---

**Algorithm 4** CUCB [8, 18]

---

1: For each arm $i$, maintain: (i) $\hat{\mu}_i$, the average of all observed outcomes from arm $i$ so far, and (ii) $T_i$, the number of observed outcomes from arm $i$ so far.

2: // Initialization
3: **for** $i = 1$ **to** $m$ **do**
4:     // Action in the $i$-th round
5:     Play a super arm $S_i$ that contains arm $i$, and update $\hat{\mu}_i$ and $T_i$.
6: **end for**

7: **for** $t = m + 1, m + 2, \ldots$ **do**
8:     // Action in the $t$-th round
9:     $\bar{\mu}_i \leftarrow \min\{\hat{\mu}_i + \sqrt{\frac{3\ln t}{2T_i}}, 1\}$     $\forall i \in [m]$
10:     Play the super arm $S_t \leftarrow \text{Oracle}(\bar{\mu})$, where $\bar{\mu} = (\bar{\mu}_1, \ldots, \bar{\mu}_m)$.
11:     Update $\hat{\mu}_i$ and $T_i$ for all $i \in S_t$.
12: **end for**

---

### A.2   Analysis of Our Algorithm in the Previous CMAB Framework

We now give an analysis of SDCB in the previous CMAB framework, following our discussion in Section 3. We consider the case in which the expected reward only depends on the means of the random variables. Namely, $r_D(S)$ only depends on $\mu_i$'s ($i \in S$), where $\mu_i$ is arm $i$'s mean outcome. In this case, we can rewrite $r_D(S)$ as $r_\mu(S)$, where $\mu = (\mu_1, \ldots, \mu_m)$ is the vector of means. Note that the offline computation oracle only needs a mean vector as input.

We no longer need the three assumptions (Assumptions 1-3) given in Section 2. In particular, we do not require independence among outcome distributions of all arms (Assumption 1). Although we cannot write $D$ as $D = D_1 \times \cdots \times D_m$, we still let $D_i$ be the outcome distribution of arm $i$. In this case, $D_i$ is the marginal distribution of $D$ in the $i$-th component.

We summarize the CUCB algorithm [8, 18] in Algorithm 4. It maintains the empirical mean $\hat{\mu}_i$ of the outcomes from each arm $i$, and stores the number of observed outcomes from arm $i$ in a variable $T_i$. In each round, it calculates an upper confidence bound (UCB) $\bar{\mu}_i$ of $\mu_i$, Then it uses the UCB vector $\bar{\mu}$ as the input to the oracle, and plays the super arm output by the oracle. In the $t$-th round ($t > m$), each UCB $\bar{\mu}_i$ has the key property that

$$\mu_i \le \bar{\mu}_i \le \mu_i + 2\sqrt{\frac{3\ln t}{2T_{i,t-1}}} \tag{22}$$

holds with high probability. (Recall that $T_{i,t-1}$ is the value of $T_i$ after $t-1$ rounds.) To see this, note that we have $|\mu_i - \hat{\mu}_i| \le \sqrt{\frac{3\ln t}{2T_{i,t-1}}}$ with high probability (by Chernoff bound), and then (22) follows from the definition of $\bar{\mu}_i$ in line 9 of Algorithm 4.

We prove that the same property as (22) also holds for SDCB. Consider a fixed $t > m$, and let $\underline{D} = \underline{D}_1 \times \cdots \times \underline{D}_m$ be the input to the oracle in the $t$-th round of SDCB. Let $\nu_i = \mathbb{E}_{Y_i \sim \underline{D}_i}[Y_i]$. We can think that SDCB uses the mean vector $\nu = (\nu_1, \ldots, \nu_m)$ as the input to the oracle used by CUCB. We now show that for each $i$, we have

$$\mu_i \le \nu_i \le \mu_i + 2\sqrt{\frac{3\ln t}{2T_{i,t-1}}} \tag{23}$$

with high probability.

To show (23), we first prove the following lemma.

**Lemma 6.** *Let $P$ and $P'$ be two distributions over $[0,1]$ with CDFs $F$ and $F'$, respectively. Consider two random variables $Y \sim P$ and $Y' \sim P'$.*

*(i) If for all $x \in [0,1]$ we have $F'(x) \le F(x)$, then we have $\mathbb{E}[Y] \le \mathbb{E}[Y']$.*

*(ii) If for all $x \in [0,1]$ we have $F(x) - F'(x) \le \Lambda$ ($\Lambda > 0$), then we have $\mathbb{E}[Y'] \le \mathbb{E}[Y] + \Lambda$.*

*Proof.* We have

$$\mathbb{E}[Y] = \int_0^1 x \, \mathrm{d}F(x) = (xF(x))\big|_0^1 - \int_0^1 F(x) \, \mathrm{d}x = 1 - \int_0^1 F(x) \, \mathrm{d}x.$$

Similarly, we have

$$\mathbb{E}[Y'] = 1 - \int_0^1 F'(x) \, \mathrm{d}x.$$

Then the lemma holds trivially. □

Now we prove (23). According to the DKW inequality, with high probability we have

$$F_i(x) - 2\sqrt{\frac{3 \ln t}{2T_{i,t-1}}} \leq \underline{F}_i(x) \leq F_i(x) \tag{24}$$

for all $i \in [m]$ and $x \in [0,1]$, where $\underline{F}_i$ is the CDF of $\underline{D}_i$ used in round $t$ of SDCB, and $F_i$ is the CDF of $D_i$. Suppose (24) holds for all $i, x$, then for any $i$, the two distributions $D_i$ and $\underline{D}_i$ satisfy the two conditions in Lemma 6, with $\Lambda = 2\sqrt{\frac{3 \ln t}{2T_{i,t-1}}}$; then from Lemma 6 we know that $\mu_i \leq \nu_i \leq \mu_i + 2\sqrt{\frac{3 \ln t}{2T_{i,t-1}}}$. Hence we have shown that (23) holds with high probability.

The fact that (23) holds with high probability means that the mean of $\underline{D}_i$ is also a UCB of $\mu_i$ with the same confidence as in CUCB. With this property, the analysis in [8, 18] can also be applied to SDCB, resulting in exactly the same regret bounds.

## B  Missing Proofs from Section 4

### B.1  Analysis of the Discretization Error

The following lemma gives an upper bound on the error due to discretization. Refer to Section 4 for the definition of the discretized distribution $\tilde{D}$.

**Lemma 7.** *For any $S \in \mathcal{F}$, we have*

$$|r_D(S) - r_{\tilde{D}}(S)| \leq \frac{CK}{s}.$$

To prove Lemma 7, we show a slightly more general lemma which gives an upper bound on the discretization error of the expectation of a Lipschitz continuous function.

**Lemma 8.** *Let $g(x)$ be a Lipschitz continuous function on $[0,1]^n$ such that for any $x, x' \in [0,1]^n$, we have $|g(x) - g(x')| \leq C\|x - x'\|_1$, where $\|x - x'\|_1 = \sum_{i=1}^n |x_i - x'_i|$. Let $P = P_1 \times \cdots \times P_n$ be a probability distribution over $[0,1]^n$. Define another distribution $\tilde{P} = \tilde{P}_1 \times \cdots \times \tilde{P}_n$ over $[0,1]^n$ as follows: each $\tilde{P}_i$ ($i \in [n]$) takes values in $\{\frac{1}{s}, \frac{2}{s}, \ldots, 1\}$, and*

$$\Pr_{\tilde{X}_i \sim \tilde{P}_i}[\tilde{X}_i = j/s] = \Pr_{X_i \sim P_i}[X_i \in I_j], \qquad j \in [s],$$

*where $I_1 = [0, \frac{1}{s}], I_2 = (\frac{1}{s}, \frac{2}{s}], \ldots, I_{s-1} = (\frac{s-2}{s}, \frac{s-1}{s}], I_s = (\frac{s-1}{s}, 1]$. Then*

$$\left|\mathbb{E}_{X \sim P}[g(X)] - \mathbb{E}_{\tilde{X} \sim \tilde{P}}[g(\tilde{X})]\right| \leq \frac{C \cdot n}{s}. \tag{25}$$

*Proof.* Throughout the proof, we consider $X = (X_1, \ldots, X_n) \sim P$ and $\tilde{X} = (\tilde{X}_1, \ldots, \tilde{X}_n) \sim \tilde{P}$.

Let $v_j = \frac{j}{s}$ ($j = 0, 1, \ldots, s$) and

$$p_{i,j} = \Pr[\tilde{X}_i = v_j] = \Pr[X_i \in I_j] \qquad i \in [n], j \in [s].$$

We prove (25) by induction on $n$.

(1) When $n = 1$, we have

$$\mathbb{E}[g(X_1)] = \sum_{j \in [s], p_{1,j} > 0} p_{1,j} \cdot \mathbb{E}\left[g(X_1)\big| X_1 \in I_j\right]. \tag{26}$$

Since $g$ is continuous, for each $j \in [s]$ such that $p_{1,j} > 0$, there exists $\xi_j \in [v_{j-1}, v_j]$ such that

$$\mathbb{E}\left[g(X_1)|X_1 \in I_j\right] = g(\xi_j)$$

From the Lipschitz continuity of $g$ we have

$$|g(v_j) - g(\xi_j)| \le C|v_j - \xi_j| \le C|v_j - v_{j-1}| = \frac{C}{s}.$$

Hence

$$
\begin{aligned}
\left|\mathbb{E}[g(X_1)] - \mathbb{E}[g(\tilde{X}_1)]\right| &= \left| \sum_{j \in [s], p_{1,j} > 0} p_{1,j} \cdot \mathbb{E}[g(X_1)|X_1 \in I_j] - \sum_{j \in [s], p_{1,j} > 0} p_{1,j} \cdot g(v_j) \right| \\
&= \left| \sum_{j \in [s], p_{1,j} > 0} p_{1,j} \cdot g(\xi_j) - \sum_{j \in [s], p_{1,j} > 0} p_{1,j} \cdot g(v_j) \right| \\
&\le \sum_{j \in [s], p_{1,j} > 0} p_{1,j} \cdot |g(\xi_j) - g(v_j)| \\
&\le \sum_{j \in [s], p_{1,j} > 0} p_{1,j} \cdot \frac{C}{s} \\
&= \frac{C}{s}.
\end{aligned}
$$

This proves (25) for $n = 1$.

(ii) Suppose (25) is correct for $n = 1, 2, \ldots, k-1$. Now we prove it for $n = k$ ($k \ge 2$).
We define two functions on $[0,1]^{k-1}$:

$$h(x_1, \ldots, x_{k-1}) = \mathbb{E}_{X_k}[g(x_1, \ldots, x_{k-1}, X_k)]$$

and

$$\tilde{h}(x_1, \ldots, x_{k-1}) = \mathbb{E}_{\tilde{X}_k}[g(x_1, \ldots, x_{k-1}, \tilde{X}_k)].$$

For any fixed $x_1, \ldots, x_{k-1} \in [0,1]$, the function $g(x_1, \ldots, x_{k-1}, x)$ on $x \in [0,1]$ is Lipschitz continuous. Therefore from the result for $n = 1$ we have

$$\left|h(x_1, \ldots, x_{k-1}) - \tilde{h}(x_1, \ldots, x_{k-1})\right| \le \frac{C}{s} \qquad \forall x_1, \ldots, x_{k-1} \in [0,1].$$

Then we have

$$\left| \mathbb{E}[g(X)] - \mathbb{E}[g(\tilde{X})] \right|$$

$$= \left| \mathbb{E}_{X_1,\ldots,X_{k-1}} \left[ \mathbb{E}[g(X)|X_1,\ldots,X_{k-1}] \right] - \mathbb{E}[g(\tilde{X})] \right|$$

$$= \left| \mathbb{E}_{X_1,\ldots,X_{k-1}} \left[ h(X_1,\ldots,X_{k-1}) \right] - \mathbb{E}[g(\tilde{X})] \right|$$

$$\leq \left| \mathbb{E}_{X_1,\ldots,X_{k-1}}[h(X_1,\ldots,X_{k-1})] - \mathbb{E}_{X_1,\ldots,X_{k-1}}[\tilde{h}(X_1,\ldots,X_{k-1})] \right|$$

$$\quad + \left| \mathbb{E}_{X_1,\ldots,X_{k-1}}[\tilde{h}(X_1,\ldots,X_{k-1})] - \mathbb{E}[g(\tilde{X})] \right|$$

$$\leq \mathbb{E}_{X_1,\ldots,X_{k-1}} \left[ \left| h(X_1,\ldots,X_{k-1}) - \tilde{h}(X_1,\ldots,X_{k-1}) \right| \right]$$

$$\quad + \left| \mathbb{E}_{X_1,\ldots,X_{k-1},\tilde{X}_k}[g(X_1,\ldots,X_{k-1},\tilde{X}_k)] - \mathbb{E}[g(\tilde{X})] \right|$$

$$\leq \mathbb{E}_{X_1,\ldots,X_{k-1}} \left[ \frac{C}{s} \right] + \left| \mathbb{E}_{\tilde{X}_k} \left[ \mathbb{E}[g(X_1,\ldots,X_{k-1},\tilde{X}_k)|\tilde{X}_k] - \mathbb{E}[g(\tilde{X}_1,\ldots,\tilde{X}_{k-1},\tilde{X}_k)|\tilde{X}_k] \right] \right|$$

$$\leq \frac{C}{s} + \mathbb{E}_{\tilde{X}_k} \left[ \left| \mathbb{E}[g(X_1,\ldots,X_{k-1},\tilde{X}_k)|\tilde{X}_k] - \mathbb{E}[g(\tilde{X}_1,\ldots,\tilde{X}_{k-1},\tilde{X}_k)|\tilde{X}_k] \right| \right]$$

$$= \frac{C}{s} + \sum_{j\in[s],p_{k,j}>0} p_{k,j} \cdot \left| \mathbb{E}[g(X_1,\ldots,X_{k-1},v_j)] - \mathbb{E}[g(\tilde{X}_1,\ldots,\tilde{X}_{k-1},v_j)] \right|.$$

$$(27)$$

For any $j \in [s]$, the function $g(x_1,\ldots,x_{k-1},v_j)$ on $(x_1,\ldots,x_{k-1}) \in [0,1]^{k-1}$ is Lipschitz continuous. Then from the induction hypothesis at $n = k-1$, we have

$$\left| \mathbb{E}[g(X_1,\ldots,X_{k-1},v_j)] - \mathbb{E}[g(\tilde{X}_1,\ldots,\tilde{X}_{k-1},v_j)] \right| \leq \frac{C(k-1)}{s} \qquad \forall j \in [s]. \qquad (28)$$

From (27) and (28) we have

$$\left| \mathbb{E}[g(X)] - \mathbb{E}[g(\tilde{X})] \right| \leq \frac{C}{s} + \sum_{j\in[s],p_{k,j}>0} p_{k,j} \cdot \frac{C(k-1)}{s}$$

$$= \frac{C}{s} + \frac{C(k-1)}{s}$$

$$= \frac{Ck}{s}.$$

This concludes the proof for $n = k$. $\qquad\qquad\square$

Now we prove Lemma 7.

*Proof of Lemma 7.* We have

$$r_D(S) = \mathbb{E}_{X\sim D}[R(X,S)] = \mathbb{E}_{X\sim D}[R_S(X_S)] = \mathbb{E}_{X_S\sim D_S}[R_S(X_S)],$$

where $X_S = (X_i)_{i\in S}$ and $D_S = (D_i)_{i\in S}$. Similarly, we have

$$r_{\tilde{D}}(S) = \mathbb{E}_{\tilde{X}_S\sim\tilde{D}_S}[R_S(\tilde{X}_S)].$$

According to Assumption 4, the function $R_S$ defined on $[0,1]^S$ is Lipschitz continuous. Then from Lemma 8 we have

$$|r_D(S) - r_{\tilde{D}}(S)| = \left| \mathbb{E}_{X_S\sim D_S}[R_S(X_S)] - \mathbb{E}_{\tilde{X}_S\sim\tilde{D}_S}[R_S(\tilde{X}_S)] \right| \leq \frac{C\cdot|S|}{s} \leq \frac{C\cdot K}{s}.$$

This completes the proof. $\qquad\qquad\square$

## B.2 Proof of Theorem 2

*Proof of Theorem 2.* Let $S^* = \mathrm{argmax}_{S\in\mathcal{F}}\{r_D(S)\}$ and $\tilde{S}^* = \mathrm{argmax}_{S\in\mathcal{F}}\{r_{\tilde{D}}(S)\}$ be the optimal super arms in problems $([m], \mathcal{F}, D, R)$ and $([m], \mathcal{F}, \tilde{D}, R)$, respectively. Suppose Algorithm 2 selects super arm $S_t$ in the $t$-th round $(1 \le t \le T)$. Then its $\alpha$-approximation regret is bounded as

$$\mathrm{Reg}^{\mathrm{Alg.\,2}}_{D,\alpha}(T)$$

$$= T \cdot \alpha \cdot r_D(S^*) - \sum_{t=1}^{T} \mathbb{E}\left[r_D(S_t)\right]$$

$$= T \cdot \alpha \left(r_D(S^*) - r_{\tilde{D}}(\tilde{S}^*)\right) + \sum_{t=1}^{T} \mathbb{E}\left[r_{\tilde{D}}(S_t) - r_D(S_t)\right] + \left(T \cdot \alpha \cdot r_{\tilde{D}}(\tilde{S}^*) - \sum_{t=1}^{T} \mathbb{E}\left[r_{\tilde{D}}(S_t)\right]\right)$$

$$\le T \cdot \alpha \left(r_D(S^*) - r_{\tilde{D}}(S^*)\right) + \sum_{t=1}^{T} \mathbb{E}\left[r_{\tilde{D}}(S_t) - r_D(S_t)\right] + \mathrm{Reg}^{\mathrm{Alg.\,1}}_{\tilde{D},\alpha}(T).$$

where the inequality is due to $r_{\tilde{D}}(\tilde{S}^*) \ge r_{\tilde{D}}(S^*)$.

Then from Lemma 7 and the distribution-independent bound in Theorem 1 we have

$$\mathrm{Reg}^{\mathrm{Alg.\,2}}_{D,\alpha}(T) \le T \cdot \alpha \cdot \frac{CK}{s} + T \cdot \frac{CK}{s} + 93M\sqrt{mKT\ln T} + \left(\frac{\pi^2}{3}+1\right)\alpha Mm$$

$$\le 2 \cdot \frac{CKT}{s} + 93M\sqrt{mKT\ln T} + \left(\frac{\pi^2}{3}+1\right)\alpha Mm \qquad (29)$$

$$\le 93M\sqrt{mKT\ln T} + 2CK\sqrt{T} + \left(\frac{\pi^2}{3}+1\right)\alpha Mm.$$

Here in the last two inequalities we have used $\alpha \le 1$ and $s = \lceil\sqrt{T}\rceil \ge \sqrt{T}$. The proof is completed.

$\square$

## B.3 Proof of Theorem 3

*Proof of Theorem 3.* Let $n = \lceil\log_2 T\rceil$. Then we have $2^{n-1} < T \le 2^n$.

If $n \le q = \lceil\log_2 m\rceil$, then $T \le 2m$ and the regret in $T$ rounds is at most $2m \cdot \alpha M$. The regret bound holds trivially.

Now we assume $n \ge q + 1$. Using Theorem 2, we have

$$\mathrm{Reg}^{\mathrm{Alg.\,3}}_{D,\alpha}(T)$$

$$\le \mathrm{Reg}^{\mathrm{Alg.\,3}}_{D,\alpha}(2^n)$$

$$= \mathrm{Reg}^{\mathrm{Alg.\,2}}_{D,\alpha}(2^q) + \sum_{k=q}^{n-1} \mathrm{Reg}^{\mathrm{Alg.\,2}}_{D,\alpha}(2^k)$$

$$\le \mathrm{Reg}^{\mathrm{Alg.\,2}}_{D,\alpha}(2m) + \sum_{k=q}^{n-1} \mathrm{Reg}^{\mathrm{Alg.\,2}}_{D,\alpha}(2^k)$$

$$\le 2m \cdot \alpha M + \sum_{k=q}^{n-1} \left(93M\sqrt{mK \cdot 2^k \ln 2^k} + 2CK\sqrt{2^k} + \left(\frac{\pi^2}{3}+1\right)\alpha Mm\right)$$

$$\le 2\alpha Mm + \left(93M\sqrt{mK\ln 2^{n-1}} + 2CK\right) \cdot \sum_{k=1}^{n-1}\sqrt{2^k} + (n-1)\cdot\left(\frac{\pi^2}{3}+1\right)\alpha Mm$$

$$\le \left(93M\sqrt{mK\ln 2^{n-1}} + 2CK\right) \cdot \frac{\sqrt{2^n}}{\sqrt{2}-1} + \left(\frac{\pi^2}{3}+3\right)(n-1)\cdot\alpha Mm$$

**Algorithm 5** `Greedy-K-MAX`

---
1: $S \leftarrow \emptyset$
2: **for** $i = 1$ **to** $K$ **do**
3:     $k \leftarrow \text{argmax}_{j \in [m] \setminus S}\, r_D(S \cup \{j\})$
4:     $S \leftarrow S \cup \{k\}$
5: **end for**
**Output:** $S$

---

$$
\leq \left(93M\sqrt{mK\ln T} + 2CK\right) \cdot \frac{\sqrt{2T}}{\sqrt{2}-1} + \left(\frac{\pi^2}{3} + 3\right)\log_2 T \cdot \alpha Mm
$$
$$
\leq 318M\sqrt{mKT\ln T} + 7CK\sqrt{T} + 10\alpha Mm\ln T. \qquad \square
$$

## C    The Offline $K$-MAX Problem

In this section, we consider the offline $K$-MAX problem. Recall that we have $m$ independent random variables $\{X_i\}_{i\in[m]}$. $X_i$ follows the discrete distribution $D_i$ with support $\{v_{i,1}, \ldots, v_{i,s_i}\} \subset [0,1]$, and $D = D_1 \times \cdots \times D_m$ is the joint distribution of $X = (X_1, \ldots, X_m)$. Let $p_{i,j} = \Pr[X_i = v_{i,j}]$. Define $r_D(S) = \mathbb{E}_{X\sim D}[\max_{i\in S} X_i]$ and $\mathsf{OPT} = \max_{S:|S|=K} r_D(S)$. Our goal is to find (in polynomial time) a subset $S \subseteq [m]$ of cardinality $K$ such that $r_D(S) \geq \alpha \cdot \mathsf{OPT}$ (for certain constant $\alpha$).

First, we show that $r_D(S)$ can be calculated in polynomial time given any $S \subseteq [m]$. Let $S = \{i_1, i_2, \ldots, i_n\}$. Note that for $X \sim D$, $\max_{i\in S} X_i$ can only take values in the set $V(S) = \bigcup_{i\in S} \text{supp}(D_i)$. For any $v \in V(S)$, we have

$$
\begin{aligned}
&\Pr_{X\sim D}\left[\max_{i\in S} X_i = v\right] \\
&= \Pr_{X\sim D}[X_{i_1} = v, X_{i_2} \leq v, \ldots, X_{i_n} \leq v] \\
&\quad + \Pr_{X\sim D}[X_{i_1} < v, X_{i_2} = v, X_{i_3} \leq v, \ldots, X_{i_n} \leq v] \\
&\quad + \cdots \\
&\quad + \Pr_{X\sim D}[X_{i_1} < v, \ldots, X_{i_{n-1}} < v, X_{i_n} = v].
\end{aligned}
\tag{30}
$$

Since $X_{i_1}, \ldots, X_{i_n}$ are mutually independent, each probability appearing in (30) can be calculated in polynomial time. Hence for any $v \in V(S)$, $\Pr_{X\sim D}[\max_{i\in S} X_i = v]$ can be calculated in polynomial time using (30). Then $r_D(S)$ can be calculated by

$$
r_D(S) = \sum_{v\in V(S)} v \cdot \Pr_{X\sim D}\left[\max_{i\in S} X_i = v\right]
$$

in polynomial time.

### C.1    $(1 - 1/e)$-Approximation

We now show that a simple greedy algorithm (Algorithm 5) can find a $(1-1/e)$-approximate solution, by proving the submodularity of $r_D(S)$. In fact, this is implied by a slightly more general result [13, Lemma 3.2]. We provide a simple and direct proof for completeness.

**Lemma 9.** *Algorithm 5 can output a subset $S$ such that $r_D(S) \geq (1 - 1/e) \cdot \mathsf{OPT}$.*

*Proof.* For any $x \in [0,1]^m$, let $f_x(S) = \max_{i\in S} x_i$ be a set function defined on $2^{[m]}$. (Define $f_x(\emptyset) = 0$.) We can verify that $f_x(S)$ is monotone and submodular:

- *Monotonicity.* For any $A \subseteq B \subseteq [m]$, we have $f_x(A) = \max_{i\in A} x_i \leq \max_{i\in B} x_i = f_x(B)$.

- *Submodularity.* For any $A \subseteq B \subseteq [m]$ and any $k \in [m] \setminus B$, there are three cases (note that $\max_{i \in A} x_i \leq \max_{i \in B} x_i$):

  (i) If $x_k \leq \max_{i \in A} x_i$, then $f_x(A \cup \{k\}) - f_x(A) = 0 = f_x(B \cup \{k\}) - f_x(B)$.

  (ii) If $\max_{i \in A} x_i < x_k \leq \max_{i \in B} x_i$, then $f_x(A \cup \{k\}) - f_x(A) = x_k - \max_{i \in A} x_i > 0 = f_x(B \cup \{k\}) - f_x(B)$.

  (iii) If $x_k > \max_{i \in B} x_i$, then $f_x(A \cup \{k\}) - f_x(A) = x_k - \max_{i \in A} x_i \geq x_k - \max_{i \in B} x_i = f_x(B \cup \{k\}) - f_x(B)$.

  Therefore, we always have $f_x(A \cup \{k\}) - f_x(A) \geq f_x(B \cup \{i\}) - f_x(B)$. The function $f_x(S)$ is submodular.

For any $S \subseteq [m]$ we have

$$r_D(S) = \sum_{j_1=1}^{s_1} \sum_{j_2=1}^{s_2} \cdots \sum_{j_m=1}^{s_m} f_{(v_{1,j_1}, \ldots, v_{m,j_m})}(S) \prod_{i=1}^{m} p_{i,j_i}.$$

Since each set function $f_{(v_{1,j_1}, \ldots, v_{m,j_m})}(S)$ is monotone and submodular, $r_D(S)$ is a convex combination of monotone submodular functions on $2^{[m]}$. Therefore, $r_D(S)$ is also a monotone submodular function. According to the classical result on submodular maximization [25], the greedy algorithm can find a $(1 - 1/e)$-approximate solution to $\max_{S \subseteq [m], |S| \leq K} \{r_D(S)\}$. $\qquad\square$

### C.2 PTAS

Now we provide a PTAS for the $K$-MAX problem. In other words, we give an algorithm which, given any fixed constant $0 < \varepsilon < 1/2$, can find a solution $S$ of cardinality $|K|$ such that $r_D(S) \geq (1 - \varepsilon) \cdot \mathsf{OPT}$ in polynomial time.

We first provide an overview of our approach, and then spell out the details later.

1. (Discretization) We first transform each $X_i$ to another discrete distribution $\tilde{X}_i$, such that all $\tilde{X}_i$'s are supported on a set of size $O(1/\varepsilon^2)$.

2. (Computing signatures) For each $X_i$, we can compute from $\tilde{X}_i$ a signature $\mathsf{Sig}(X_i)$ which is a vector of size $O(1/\varepsilon^2)$. For a set $S$, we define its signature $\mathsf{Sig}(S)$ to be $\sum_{i \in S} \mathsf{Sig}(X_i)$. We show that if two sets $S_1$ and $S_2$ have the same signature, their objective values are close (Lemma 12).

3. (Enumerating signatures) We enumerate all possible signatures (there are polynomial number of them when treating $\varepsilon$ as a constant) and try to find the one which is the signature of a set of size $K$, and the objective value is maximized.

#### C.2.1 Discretization

We first describe the discretization step. We say that a random variable $X$ follows the Bernoulli distribution $B(v, q)$ if $X$ takes value $v$ with probability $q$ and value $0$ with probability $1 - q$. For any discrete distribution, we can rewrite it as the maximum of a set of Bernoulli distributions.

**Definition 1.** *Let $X$ be a discrete random variable with support $\{v_1, v_2, \ldots, v_s\}$ ($v_1 < v_2 < \cdots < v_s$) and $\Pr[X = v_j] = p_j$. We define a set of independent Bernoulli random variables $\{Z_j\}_{j \in [s]}$ as*

$$Z_j \sim B\left(v_j, \frac{p_j}{\sum_{j' \leq j} p_{j'}}\right).$$

*We call $\{Z_j\}$ the Bernoulli decomposition of $X_i$.*

**Lemma 10.** *For a discrete distribution $X$ and its Bernoulli decomposition $\{Z_j\}$, $\max_j \{Z_j\}$ has the same distribution with $X$.*

*Proof.* We can easily see the following:

$$\Pr[\max_j \{Z_j\} = v_i] = \Pr[Z_i = v_i] \prod_{i' > i} \Pr[Z_{i'} = 0]$$

---

**Algorithm 6** Discretization

---
1: We first run `Greedy-K-MAX` to obtain a solution $S_G$ and let $\mathsf{W} = r_D(S_G)$.
2: **for** $i = 1$ **to** $m$ **do**
3:     Compute the Bernoulli decomposition $\{Z_{i,j}\}_j$ of $X_i$.
4:     **for all** $Z_{i,j}$ **do**
5:         Create another Bernoulli variable $\tilde{Z}_{i,j}$ as follows:
6:         **if** $v_{i,j} > \mathsf{W}/\varepsilon$ **then**
7:             Let $\tilde{Z}_{i,j} \sim B\left(\frac{\mathsf{W}}{\varepsilon}, \mathbb{E}[Z_{i,j}]\frac{\varepsilon}{\mathsf{W}}\right)$ (Case 1)
8:         **else**
9:             Let $\tilde{Z}_{i,j} = \lfloor \frac{Z_{i,j}}{\varepsilon\mathsf{W}} \rfloor \varepsilon\mathsf{W}$ (Case 2)
10:         **end if**
11:     **end for**
12:     Let $\tilde{X}_i = \max_j\{\tilde{Z}_{ij}\}$
13: **end for**

---

$$= \frac{p_i}{\sum_{i' \leq i} p_{i'}} \prod_{h > i} \left(1 - \frac{p_h}{\sum_{h' \leq h} p_{h'}}\right)$$

$$= \frac{p_i}{\sum_{i' \leq i} p_{i'}} \prod_{h > i} \frac{\sum_{h' \leq h-1} p_{h'}}{\sum_{h' \leq h} p_{h'}} = p_i.$$

Hence, $\Pr[\max_j\{Z_j\} = v_i] = \Pr[X = v_i]$ for all $i \in [s]$. $\qquad\square$

Now, we describe how to construct the discretization $\tilde{X}_i$ of $X_i$ for all $i \in [m]$. The pseudocode can be found in Algorithm 6. We first run `Greedy-K-MAX` to obtain a solution $S_G$. Let $\mathsf{W} = r_D(S_G)$. By Lemma 9, we know that $\mathsf{W} \geq (1 - 1/e)\mathsf{OPT}$. Then we compute the Bernoulli decomposition $\{Z_{i,j}\}_j$ of $X_i$. For each $Z_{i,j}$, we create another Bernoulli variable $\tilde{Z}_{i,j}$ as follows: Recall that $v_{i,j}$ is the nonzero possible value of $Z_{ij}$. We distinguish two cases. Case 1: If $v_{i,j} > \mathsf{W}/\varepsilon$, then we let $\tilde{Z}_{i,j} \sim B\left(\frac{\mathsf{W}}{\varepsilon}, \mathbb{E}[Z_{i,j}]\frac{\varepsilon}{\mathsf{W}}\right)$. It is easy to see that $\mathbb{E}[\tilde{Z}_{ij}] = \mathbb{E}[Z_{ij}]$. Case 2: If $v_{i,j} \leq \mathsf{W}/\varepsilon$, then we let $\tilde{Z}_{i,j} = \lfloor \frac{Z_{i,j}}{\varepsilon\mathsf{W}} \rfloor \varepsilon\mathsf{W}$. We note that more than one $\tilde{Z}_{ij}$'s may have the same support, and all $\tilde{Z}_{ij}$'s are supported on $\mathsf{DS} = \{0, \varepsilon\mathsf{W}, 2\varepsilon\mathsf{W}, \ldots, \mathsf{W}/\varepsilon\}$. Finally, we let $\tilde{X}_i = \max_j\{\tilde{Z}_{ij}\}$, which is the discretization of $X_i$. Since $\tilde{X}_i$ is the maximum of a set of Bernoulli distributions, it is also a discrete distribution supported on $\mathsf{DS}$. We can easily compute $\Pr[\tilde{X}_i = v]$ for any $v \in \mathsf{DS}$.

Now, we show that the discretization only incurs a small loss in the objective value. The key is to show that we do not lose much in the transformation from $Z_{i,j}$'s to $\tilde{Z}_{i,j}$'s. We prove a slightly more general lemma as follows.

**Lemma 11.** *Consider any set of Bernoulli variables $\{Z_i \sim B(a_i, p_i)\}_{1 \leq i \leq n}$. Assume that $\mathbb{E}[\max_{i \in [n]} Z_i] < c\mathsf{W}$, where $c$ is a constant such that $c\varepsilon < 1/2$. For each $Z_i$, we create a Bernoulli variable $\tilde{Z}_i$ in the same way as Algorithm 6. Then the following holds:*

$$\mathbb{E}[\max Z_i] \geq \mathbb{E}[\max \tilde{Z}_i] \geq \mathbb{E}[\max Z_i] - (2c + 1)\varepsilon\mathsf{W}.$$

*Proof.* Assume $a_1$ is the largest among all $a_i$'s.

If $a_1 < \mathsf{W}/\varepsilon$, all $\tilde{Z}_i$ are created in Case 2. In this case, it is obvious to have that

$$\mathbb{E}[\max Z_i] \geq \mathbb{E}[\max \tilde{Z}_i] \geq \mathbb{E}[\max Z_i] - \varepsilon\mathsf{W}.$$

If $a_1 \geq \mathsf{W}/\varepsilon$, the proof is slightly more complicated. Let $L = \{i \mid a_i \geq \mathsf{W}/\varepsilon\}$. We prove by induction on $n$ (i.e., the number of the variables) the following more general claim:

$$\mathbb{E}[\max Z_i] \geq \mathbb{E}[\max \tilde{Z}_i] \geq \mathbb{E}[\max Z_i] - \varepsilon\mathsf{W} - c\sum_{i \in L} \varepsilon a_i p_i. \tag{31}$$

Consider the base case $n = 1$. The lemma holds immediately in Case 1 as $\mathbb{E}[Z_1] = \mathbb{E}[\tilde{Z}_1]$.

Assuming the lemma is true for $n = k$, we show it also holds for $n = k + 1$. Recall we have $\tilde{Z}_1 \sim B(\frac{\mathsf{W}}{\varepsilon}, \varepsilon\mathbb{E}[Z_1]/\mathsf{W})$. Thus

$$\mathbb{E}[\max_{i \geq 1} Z_i] - \mathbb{E}[\max_{i \geq 1} \tilde{Z}_i] = a_1 p_1 + (1 - p_1)\mathbb{E}[\max_{i \geq 2} Z_i] - a_1 p_1 - (1 - \varepsilon\mathbb{E}[Z_1]/\mathsf{W})\mathbb{E}[\max_{i \geq 2} \tilde{Z}_i]$$

$$\geq (1 - p_1)\mathbb{E}[\max_{i \geq 2} \tilde{Z}_i] - (1 - \varepsilon\mathbb{E}[Z_1]/\mathsf{W})\mathbb{E}[\max_{i \geq 2} \tilde{Z}_i]$$

$$= (\varepsilon a_1 p_1/\mathsf{W} - p_1)\mathbb{E}[\max_{i \geq 2} \tilde{Z}_i] \geq 0,$$

where the first inequality follows from the induction hypothesis and the last from $a_1 \geq \mathsf{W}/\varepsilon$. The other direction can be seen as follows:

$$\mathbb{E}[\max_{i \geq 1} \tilde{Z}_i] - \mathbb{E}[\max_{i \geq 1} Z_i] = a_1 p_1 + (1 - \varepsilon\mathbb{E}[Z_1]/\mathsf{W})\mathbb{E}[\max_{i \geq 2} \tilde{Z}_i] - (a_1 p_1 + (1 - p_1)\mathbb{E}[\max_{i \geq 2} Z_i])$$

$$\geq (1 - \varepsilon\mathbb{E}[Z_1]/\mathsf{W})\mathbb{E}[\max_{i \geq 2} Z_i] - (1 - p_1)\mathbb{E}[\max_{i \geq 2} Z_i] - \varepsilon\mathsf{W} - c\sum_{i \in L \setminus \{1\}} \varepsilon a_i p_i$$

$$\geq (-\varepsilon\mathbb{E}[Z_1]/\mathsf{W})\mathbb{E}[\max_{i \geq 2} Z_i] - \varepsilon\mathsf{W} - c\sum_{i \in L \setminus \{1\}} \varepsilon a_i p_i$$

$$\geq -\varepsilon\mathsf{W} - c\sum_{i \in L} \varepsilon a_i p_i,$$

where the last inequality holds since $\mathbb{E}[\max_{i \geq 2} Z_i] \leq c\mathsf{W}$. This finishes the proof of (31).

Now, we show that $\sum_{i \in L} a_i p_i \leq 2\mathsf{W}$. This can be seen as follows. First, we can see from Markov inequality that

$$\Pr[\max Z_i > \mathsf{W}/\varepsilon] \leq c\varepsilon.$$

Equivalently, we have $\prod_{i \in L}(1 - p_i) \geq 1 - c\varepsilon$. Then, we can see that

$$\mathsf{W} \geq \sum_{i \in L} a_i \prod_{j < i}(1 - p_j)p_i \geq (1 - c\varepsilon)\sum_{i \in L} a_i p_i \geq \frac{1}{2}\sum_{i \in L} a_i p_i.$$

Plugging this into (31), we prove the lemma. $\qquad\square$

**Corollary 1.** *For any set $S \subseteq [m]$, suppose $\mathbb{E}[\max_{i \in S} X_i] < c\mathsf{W}$, where $c$ is a constant such that $c\varepsilon < 1/2$. Then the following holds:*

$$\mathbb{E}[\max_{i \in S} X_i] \geq \mathbb{E}[\max_{i \in S} \tilde{X}_i] \geq \mathbb{E}[\max_{i \in S} X_i] - (2c + 1)\varepsilon\mathsf{W}.$$

### C.2.2 Signatures

For each $X_i$, we have created its discretization $\tilde{X}_i = \max_j\{\tilde{Z}_{ij}\}$. Since $\tilde{X}_i$ is a discrete distribution, we can define its Bernoulli decomposition $\{Y_{ij}\}_{j \in [h]}$ where $h = |\mathsf{DS}|$. Suppose $Y_{ij} \sim B(j\varepsilon\mathsf{W}, q_{ij})$. Now, we define the signature of $X_i$ to be the vector $\mathsf{Sig}(X_i) = (\mathsf{Sig}(X_i)_1, \ldots, \mathsf{Sig}(X_i)_h)$ where

$$\mathsf{Sig}(X_i)_j = \min\left(\left\lfloor \frac{-\ln(1 - q_{ij})}{\varepsilon^4/m} \right\rfloor, \left\lfloor \frac{\ln(1/\varepsilon^4)}{\varepsilon^4/m} \right\rfloor\right) \cdot \frac{\varepsilon^4}{m} \qquad j \in [h].$$

For any set $S$, define its signature to be

$$\mathsf{Sig}(S) = \sum_{i \in S} \mathsf{Sig}(X_i).$$

Define the set $\mathsf{SG}$ of *signature vectors* to be all nonnegative $h$-dimensional vectors, where each coordinate is an integer multiple of $\varepsilon^4/m$ and at most $m\ln(1/\varepsilon^4)$. Clearly, the size of $\mathsf{SG}$ is $O\left(\left(m\varepsilon^{-4}\log(h/\varepsilon^2)\right)^{h-1}\right) = \tilde{O}(m^{O(1/\varepsilon^2)})$, which is polynomial for any fixed constant $\varepsilon > 0$ (recall $h = |\mathsf{DS}| = O(1/\varepsilon^2)$).

Now, we prove the following crucial lemma.

**Lemma 12.** *Consider two sets $S_1$ and $S_2$. If $\mathsf{Sig}(S_1) = \mathsf{Sig}(S_2)$, the following holds:*

$$\left| \mathbb{E}[\max_{i \in S_1} \tilde{X}_i] - \mathbb{E}[\max_{i \in S_2} \tilde{X}_i] \right| \leq O(\varepsilon)\mathsf{W}.$$

*Proof.* Suppose $\{Y_{ij}\}_{j \in [h]}$ is the Bernoulli decomposition of $\tilde{X}_i$. For any set $S$, we define $Y_k(S) = \max_{i \in S} Y_{ik}$ (it is the max of a set of Bernoulli distributions). It is not hard to see that $Y_k(S)$ has a Bernoulli distribution $B(k\varepsilon\mathsf{W}, p_k(S))$ with $p_k(S) = 1 - \prod_{i \in S}(1 - q_{ik})$. As $\mathsf{Sig}(S_1) = \mathsf{Sig}(S_2)$, we have that

$$|p_k(S_1) - p_k(S_2)| = |\prod_{i \in S_1}(1 - q_{ik}) - \prod_{i \in S_2}(1 - q_{ik})|$$

$$= \left| \exp\left( \sum_{i \in S_1} \ln(1 - q_{ik}) \right) - \exp\left( \sum_{i \in S_2} \ln(1 - q_{ik}) \right) \right|$$

$$\leq 2\varepsilon^4 \qquad \forall k \in [h].$$

Noticing $\max_{i \in S} \tilde{X}_i = \max_k Y_k(S)$, we have that

$$\left| \mathbb{E}[\max_{i \in S_1} \tilde{X}_i] - \mathbb{E}[\max_{i \in S_2} \tilde{X}_i] \right| = \left| \mathbb{E}[\max_k Y_k(S_1)] - \mathbb{E}[\max_k Y_k(S_2)] \right|$$

$$\leq \frac{\mathsf{W}}{\varepsilon}\left( \sum_k |p_k(S_1) - p_k(S_2)| \right)$$

$$\leq 4h\varepsilon^3\mathsf{W} = O(\varepsilon)\mathsf{W}$$

where the first inequality follows from Lemma 1. $\qquad\square$

For any signature vector $\mathsf{sg}$, we associate to it a set of random variables $\{B_k \sim B(k\varepsilon\mathsf{W}, 1 - e^{-\mathsf{sg}_k})\}_{k=1}^h$.[8] Define the value of $\mathsf{sg}$ to be $\mathsf{Val}(\mathsf{sg}) = \mathbb{E}[\max_{k \in [h]} B_k]$.

**Corollary 2.** *For any feasible set $S$ with $\mathsf{Sig}(S) = \mathsf{sg}$, $|\mathbb{E}[\max_{i \in S} \tilde{X}_i] - \mathsf{Val}(\mathsf{sg})| \leq O(\varepsilon)\mathsf{W}$. Moreover, combining with Corollary 1, we have that $|\mathbb{E}[\max_{i \in S} X_i] - \mathsf{Val}(\mathsf{sg})| \leq O(\varepsilon)\mathsf{W}$.*

### C.2.3 Enumerating Signatures

Our algorithm enumerates all signature vectors $\mathsf{sg}$ in $\mathsf{SG}$. For each $\mathsf{sg}$, we check if we can find a set $S$ of size $K$ such that $\mathsf{Sig}(S) = \mathsf{sg}$. This can be done by a standard dynamic program in $\tilde{O}(m^{O(1/\varepsilon^2)})$ time as follows: We use Boolean variable $R[i][j][\mathsf{sg}']$ to represent whether signature vector $\mathsf{sg}' \in \mathsf{SG}$ can be dominated by $i$ variables in set $\{X_1, \ldots, X_j\}$. The dynamic programming recursion is

$$R[i][j][\mathsf{sg}'] = R[i][j-1][\mathsf{sg}'] \wedge R[i-1][j-1][\mathsf{sg}' - \mathsf{Sig}(X_j)].$$

If the answer is yes (i.e., we can find such $S$), we say $\mathsf{sg}$ is a feasible signature vector and $S$ is a candidate set. Finally, we pick the candidate set with maximum $r_D(S)$ and output the set. The pseudocode can be found in Algorithm 7.

Now, we are ready to prove Theorem 4 by showing Algorithm 7 is a PTAS for the $K$-MAX problem.

*Proof of Theorem 4.* Suppose $S^*$ is the optimal solution and $\mathsf{sg}^*$ is the signature of $S^*$. By Corollary 2, we have that $|\mathsf{OPT} - \mathsf{Val}(\mathsf{sg}^*)| \leq O(\varepsilon)\mathsf{W}$.

When Algorithm 7 is enumerating $\mathsf{sg}^*$, it can find a set $S$ such that $\mathsf{Sig}(S) = \mathsf{sg}^*$ (there exists at least one such set since $S^*$ is one). Therefore, we can see that

$$|\mathbb{E}[\max_{i \in S} X_i] - \mathbb{E}[\max_{i \in S^*} X_i]| \leq |\mathsf{Val}(\mathsf{sg}^*) - \max_{i \in S} X_i| + |\mathsf{Val}(\mathsf{sg}^*) - \mathbb{E}[\max_{i \in S^*} X_i]| \leq O(\varepsilon)\mathsf{W}.$$

Let $U$ be the output of Algorithm 7. Since $\mathsf{W} \geq (1 - 1/e)\mathsf{OPT}$, we have $r_D(U) \geq r_D(S) = \mathbb{E}[\max_{i \in S} X_i] \geq (1 - O(\varepsilon))\mathsf{OPT}$.

The running time of the algorithm is polynomial for a fixed constant $\varepsilon > 0$, since the number of signature vectors is polynomial and the dynamic program in each iteration also runs in polynomial time. Hence, we have a PTAS for the $K$-MAX problem. $\qquad\square$

**Algorithm 7** `PTAS-K-MAX`
---
1: $U \leftarrow \emptyset$
2: **for all** signature vector $\mathsf{sg} \in \mathsf{SG}$ **do**
3:     Find a set $S$ such that $|S| = K$ and $\mathsf{Sig}(S) = \mathsf{sg}$
4:     **if** $r_D(S) > r_D(U)$ **then**
5:        $U \leftarrow S$
6:     **end if**
7: **end for**
**Output:** $U$
---

---
**Algorithm 8** Online Submodular Maximization [26]
---
1: Let $\mathcal{A}_1, \mathcal{A}_2, \ldots, \mathcal{A}_K$ be $K$ instances of `Exp3`
2: **for** $t = 1, 2, \ldots$ **do**
3:     // Action in the $t$-th round
4:     **for** $i = 1$ **to** $K$ **do**
5:        Use $\mathcal{A}_i$ to select an arm $a_{t,i} \in [m]$
6:     **end for**
7:     Play the super arm $S_t \leftarrow \bigcup_{i=1}^{K} \{a_{t,i}\}$
8:     **for** $i = 1$ **to** $K$ **do**
9:        Feed back $f_t(\bigcup_{j=1}^{i} \{a_{t,j}\}) - f_t(\bigcup_{j=1}^{i-1} \{a_{t,j}\})$ as the payoff $\mathcal{A}_i$ receives for choosing $a_{t,i}$
10:     **end for**
11: **end for**
---

**Remark.** In fact, Theorem 4 can be generalized in the following way: instead of the cardinality constraint $|S| \le K$, we can have more general combinatorial constraint on the feasible set $S$. As long as we can execute line 3 in Algorithm 7 in polynomial time, the analysis wound be the same. Using the same trick as in [20], we can extend the dynamic program to a more general class of combinatorial constraints where there is a pseudo-polynomial time for the exact version[9] of the deterministic version of the corresponding problem. The class of constraints includes $s$-$t$ simple paths, knapsacks, spanning trees, matchings, etc.

## D   Empirical Comparison between the SDCB Algorithm and Online Submodular Maximization on the $K$-MAX Problem

We perform experiments to compare the SDCB algorithm with the online submodular maximization algorithm in [26], on the $K$-MAX problem.

**Online Submodular Maximization.** First we briefly describe the online submodular maximization problem considered in [26] and the algorithm therein. At the beginning, an oblivious adversary sets a sequence of submodular functions $f_1, f_2, \ldots, f_T$ on $2^{[m]}$, where $f_t$ will be used to determine the reward in the $t$-th round. In the $t$-th round, if the player selects a feasible super arm $S_t$, the reward will be $f_t(S_t)$. This model covers the $K$-MAX problem as an instance: suppose $X^{(t)} = (X_1^{(t)}, \ldots, X_m^{(t)}) \sim D$ is the outcome vector sampled in the $t$-th round, then the function $f_t(S) = \max_{i \in S} X_i^{(t)}$ is submodular and will determine the reward in the $t$-th round. We summarize the algorithm in Algorithm 8. It uses $K$ copies of the `Exp3` algorithm (see [3] for an introduction). For the $K$-MAX problem, Algorithm 8 achieves an $O(K\sqrt{mT \log m})$ upper bound on the $(1 - 1/e)$-approximation regret.

**Setup.** We set $m = 9$ and $K = 3$, i.e., there are 9 arms in total and it is allowed to select at most 3 arms in each round. We compare the performance of SDCB/`Lazy-SDCB` and the online

submodular maximization algorithm on four different distributions. Here we use the greedy algorithm `Greedy-K-MAX` (Algorithm 5) as the offline oracle.

Let $X_i \sim D_i$ $(i = 1, \ldots, 9)$. We consider the following distributions. For all of them, the optimal super arm is $S^* = \{1, 2, 3\}$.

- Distribution 1: All $D_i$'s have the same support $\{0, 0.2, 0.4, 0.6, 0.8, 1\}$.
  For $i \in \{1, 2, 3\}$, $\Pr[X_i = 0] = \Pr[X_i = 0.2] = \Pr[X_i = 0.4] = \Pr[X_i = 0.6] = \Pr[X_i = 0.8] = 0.1$ and $\Pr[X_i = 1] = 0.5$.
  For $i \in \{4, 5, 6, \ldots, 9\}$, $\Pr[X_i = 0] = 0.5$ and $\Pr[X_i = 0.2] = \Pr[X_i = 0.4] = \Pr[X_i = 0.6] = \Pr[X_i = 0.8] = \Pr[X_i = 1] = 0.1$.

- Distribution 2: All $D_i$'s have the same support $\{0, 0.2, 0.4, 0.6, 0.8, 1\}$.
  For $i \in \{1, 2, 3\}$, $\Pr[X_i = 0] = \Pr[X_i = 0.2] = \Pr[X_i = 0.4] = \Pr[X_i = 0.6] = \Pr[X_i = 0.8] = 0.1$ and $\Pr[X_i = 1] = 0.5$.
  For $i \in \{4, 5, 6, \ldots, 9\}$, $\Pr[X_i = 0] = \Pr[X_i = 0.2] = \Pr[X_i = 0.4] = \Pr[X_i = 0.6] = \Pr[X_i = 0.8] = 0.12$ and $\Pr[X_i = 1] = 0.4$.

- Distribution 3: All $D_i$'s have the same support $\{0, 0.2, 0.4, 0.6, 0.8, 1\}$.
  For $i \in \{1, 2, 3\}$, $\Pr[X_i = 0] = \Pr[X_i = 0.2] = \Pr[X_i = 0.4] = \Pr[X_i = 0.6] = \Pr[X_i = 0.8] = 0.1$ and $\Pr[X_i = 1] = 0.5$.
  For $i \in \{4, 5, 6\}$, $\Pr[X_i = 0] = \Pr[X_i = 0.2] = \Pr[X_i = 0.4] = \Pr[X_i = 0.6] = \Pr[X_i = 0.8] = 0.12$ and $\Pr[X_i = 1] = 0.4$.
  For $i \in \{7, 8, 9\}$, $\Pr[X_i = 0] = \Pr[X_i = 0.2] = \Pr[X_i = 0.4] = \Pr[X_i = 0.6] = \Pr[X_i = 0.8] = 0.16$ and $\Pr[X_i = 1] = 0.2$.

- Distribution 4: All $D_i$'s are continuous distributions on $[0, 1]$.
  For $i \in \{1, 2, 3\}$, $D_i$ is the uniform distribution on $[0, 1]$.
  For $i \in \{4, 5, 6, \ldots, 9\}$, the probability density function (PDF) of $X_i$ is

$$f(x) = \begin{cases} 1.2 & x \in [0, 0.5], \\ 0.8 & x \in (0.5, 1]. \end{cases}$$

These distributions represent several different scenarios. Distribution 1 is relatively "easy" because the suboptimal arms 4-9's distribution is far away from arms 1-3's distribution, whereas distribution 2 is "hard" since the distribution of arms 4-9 is close to the distribution of arms 1-3. In distribution 3, the distribution of arms 4-6 is close to the distribution of arms 1-3's, while arms 7-9's distribution is further away. Distribution 4 is an example of a group of continuous distributions for which `Lazy-SDCB` is more efficient than `SDCB`.

We use `SDCB` for distributions 1-3, and `Lazy-SDCB` (with known time horizon) for distribution 4. Figure 1 shows the regrets of both `SDCB` and the online submodular maximization algorithm. We plot the 1-approximation regrets instead of the $(1 - 1/e)$-approximation regrets, since the greedy oracle usually performs much better than its $(1 - 1/e)$-approximation guarantee. We can see from Figure 1 that our algorithms achieve much lower regrets in all examples.

(a) Distribution 1

(b) Distribution 2

(c) Distribution 3

(d) Distribution 4

Figure 1: Regrets of SDCB/Lazy-SDCB and Algorithm 8 on the $K$-MAX problem, for distributions 1-4. The regrets are averaged over 20 independent runs.