[Reviews · NeurIPS 2016]

Reviewer 1

Summary

The paper studies a generalization of the stochastic combinatorial multi-armed bandits problem. In this generalization, the reward of each super-arm is a general function of the reward of the constituent arms. There are a few assumptions in the model that seem reasonable. The result is a generalization of the UCB-style algorithm for this setting.

Qualitative Assessment

This is an interesting and useful generalization of the CMAB framework and the results are technically non-trivial. The algorithm requires maintaining an estimate of the distribution of reward for each arm (and not just mean reward of each arm), and this makes the result different from other generalizations of the MAB framework that I've previously seen.

Confidence in this Review

2-Confident (read it all; understood it all reasonably well)


Reviewer 2

Summary

The paper deals with the problem of combinatorial multi armed bandits (CMAB). Here, there are m arms corresponding to m distributions. In each round the player picks a subset of the arms and receives a reward having a value of f(x_1,…,x_k) with f being a fixed known function, and the x’s being the realizations of random variables drawn from the appropriate distributions. Most previous papers discussing CMAB assume a linear function f, which in particular, allows the expected value of the function to be an function of the expected values of the corresponding distribution.

Qualitative Assessment

The paper deals with the problem of combinatorial multi armed bandits (CMAB). Here, there are m arms corresponding to m distributions. In each round the player picks a subset of the arms and receives a reward having a value of f(x_1,…,x_k) with f being a fixed known function, and the x’s being the realizations of random variables drawn from the appropriate distributions. Most previous papers discussing CMAB assume a linear function f, which in particular, allows the expected value of the function to be an function of the expected values of the corresponding distribution. This in turn means that previous results estimate only the mean of these variables - something that is not enough when dealing with non-linear functions. One previous result deals with CMAB with non-linear function but it assumes that the distributions come from a finite parametric family and the techniques used there are completely different than those used in the paper. A second, somewhat unrelated result given in the paper is a PTAS for the max-K problem, where the objective is: Given m distributions and a number k \leq m, find a subset of k distributions maximizing the expected max-value. The previously known methods provide a constant approximation factor. This result is interesting, but is unrelated in the sense that it deals with an offline problem and uses different techniques. Additionally, on its own I would argue that it is simply out of scope for a machine learning conference. This being said, the same cannot be said about the CMAB problem. In terms of the motivation for the problem, I’m convinced that there is a need for algorithms for CMAB that can handle non-linear functions, and that there is room for improvement (that is given here) over the existing results. In this sense the paper is well motivated. As for the theoretical contribution: The final result for the CMAB problem is not trivial yet not very surprising. The authors estimate the CDF of each of the distributions, or of a discretized distribution if it is not discrete to begin with. They use the LCB of the CDF in order to infer a UCB for the value of each set. The analysis is standard given the existing techniques, yet requires a careful consideration to details. The main weakness in the algorithm is the handling of the discretization. It seems that two improvements are somewhat easily achievable: First, there should probably be a way to obtain instance dependent bounds for the continuous setting. It seems that by taking a confidence bound of size \sqrt{log(st)/T_{i,t}} rather than \sqrt{log(t)/T_{i,t}}, one can get a logarithmic dependence on s, rather than polynomial, which may solve this issue. If that doesn’t work, the paper should benefit from an explanation for why that doesn’t work. Second, it seems that the discretization should be adaptive to the data. Otherwise, the running time and memory are dependent of the time horizon in cases where they do not have to. Overall, the paper is well written and motivated. Its results, though having room for improvement are non-trivial and deserve publication. Minor comments: - Where else was the k-max problem discussed? Please provide a citation for this.

Confidence in this Review

2-Confident (read it all; understood it all reasonably well)


Reviewer 3

Summary

The paper consider the stochastic combinatorial MAB problem with semi-bandit feedback. The major difference from previous work is that the expected reward earned in pulling a super-arm is a function on the distributions associated with the underlying basic arms. In existing literature, most paper assume that the expected reward is a function on the means of the distributions associated with the basic arms. By constructing UCBs on the cdfs of the distributions (i.e. stochastic dominance), they propose a gap dependence O(log T) regret bound and a gap independent O(sqrt(T)) regret bound. Other crucial ingredients are

Qualitative Assessment

The paper is clearly written. By generalizing the work of Chen, Wang, Yuan and Wang (2016), combinatorial MAB problems with more complex objectives, such as K-max, could be considered. Despite the generalization, the design and analysis of the algorithm appears to be very similar to the paper by Chen, Wang, Yuan and Wang (2016). The essential novelty in the paper is the construction of a random variable Y that stochastically dominates an underlying random variable X while keeping ||Y - X|| as small as possible, given a sequence of samples of X arriving online. The analysis in the manuscript seems to follow the framework in Chen et al; the only essential difference is that instead of constructing UCBs for the means of X_i (X_i is the r.v. associated with basic arm i), now it needs to construct a random variable (say Y_i) that stochastic dominates X_i, in the sense that P[Y_i\leq x]\leq P[X_i\leq x] for all x. The latter task is achieved by suitably adjusting the support of the random variable ([0,1] in this case), and then construct LCBs on each of the P[X_i\leq x]. The latter is clearly possible, since we observe X_i every time we pull a super arm containing i, which allows us to observe {X_i\leq x} for all x. While this proposed construction is interesting, it still seems to be standard in the MAB literature. Finally, given such construct for all basic arms i\in [m], it can then be argued that similar upper bounds holds for all super arms, by the assumptions of Lipschitz continuity and monotonicity. To sum up, it appears to me that the new framework enables us to solve a larger family of combinatorial MAB problems, but the techniques behind appear similar to those in the existing literature.

Confidence in this Review

2-Confident (read it all; understood it all reasonably well)


Reviewer 4

Summary

Authors addressed the problem of combinatorial multi-armed bandit (MAB) which is a generalization of classical MAB where the learner selects a feasible subset of (at most K) arms from a given set of m arms, referred as super arm, and the observe reward which is a general non-linear function of the individual rewards of each selected arms and gave efficient algorithms for some approximate notion of regret.

Qualitative Assessment

1. The novelty of the proposed algorithm did not come out properly- the concept of keeping upper/lower confidence bounds for arms reward distribution are already prevalent in MAB literature, also to predict the super arms author use an \alpha-approximation oracle at each iteration, which might itself be computationally inefficient. 2. Authors did not provide any run time guarantee for the proposed algorithms, as mentioned in the previous comment, authors use an \alpha-approximation oracle in all versions of their proposed algorithm which essentially optimize regret over a set of subsets of conspiratorially large cardinality, its nowhere mentioned under what assumptions a computationally efficient oracle is available (only one example is given for K-MAX problem). 3. Authors should have provided experimental results to support the theoretical guarantees, and also compare their methods with other baselines to compare the regret performance as well as runtime performance. 4. The algorithms are based on 4 main assumption, among these the first one itself might not be practically realistic specially when one consider the reward function of super arms are some function of rewards of each individual arms, the rewards of each base arms need not be independent, e.g. the auction problem. 5. The idea of discretizing the general reward distributions seems computationally inefficient since s might be large on certain setting, especially when T is large.

Confidence in this Review

2-Confident (read it all; understood it all reasonably well)


Reviewer 5

Summary

This paper extends the analysis of combinatorial multi-armed bandits (CMAB) to nonlinear reward functions, where the expected reward of any fixed superarm need not depend linearly on the individual expected rewards of the constituent arms. The author(s) propose an algorithm called stochastically dominant confidence bound (SDCB), which in each iteration plays a superarm and updates lower confidence bounds on the reward distributions of individual arms. The regret bounds are derived under several settings: (i) distribution dependent and distribution independent bounds (ii) finite-support and general reward distributions. The algorithms and results are applied to the K-MAX and utility maximization problems. In particular, the author(s) also propose a PTAS for offline K-MAX and then show that by using the offline oracle, the proposed CMAB algorithm can achieve (1-eps)-approximate regret in the online setting.

Qualitative Assessment

This paper is well motivated, clearly written and has strong technical contributions. The potential applications of CMAB with nonlinear reward functions are many, and several were well explored in this paper. For example, the author(s) made substantial contributions to the analysis the K-MAX bandit problem, proposing the first PTAS for the offline problem and showing that the proposed CMAB methods can achieve tight approximate regret bounds in the online setting by using the offline oracle. However, much of the work on K-MAX in this paper were relegated to the Appendix; I feel that this makes it too long and could obscure its visibility to a wider audience. Perhaps it is best presented in a longer paper or even in separate ones. Nonetheless, I recommend the publication of this paper.

Confidence in this Review

2-Confident (read it all; understood it all reasonably well)


Reviewer 6

Summary

Paper 903 generalizes the well-studied problem of stochastic combinatorial multi-arm bandit (CMAB) to the setting where the reward function is a Lipschitz - though not necessarily linear - function of the random realization of the arms. At each round, the algorithm selects a subset S - called a 'super arm - of m base arms, and receives a reward R() which is a Lipschitz and monotone function of the rewards of all individual arms. Like linear CMAB, the subset S is relies in some restricted class of subsets (say, [m]-choose-K), and the rewards of each arm assumed to be independent, and identically distributed over rounds. The authors introduce an algorithm which obtains O(log T)-instance dependent and O(sqrt(T))-worst-case simple regret with respect to alpha * the reward of the best of simple arm S^*, which makes exactly one call to an offline alpha-approximation oracle per round. The paper establishes upper bounds even when each individual arm has an arbitrary, compactly supported distribution, which qualitatively match those for linear CMAB. Moreover, 903 introduces a PTAS for the offline oracle when the reward function is the max of all the arms selected - which they denote the K-MAX problem.

Qualitative Assessment

All in all, the paper is well written, the proofs are clear, and the results are compelling. Though the algorithm is a natural extension of the linear CMAB algorithm in [Kveton et. al. 15], its key innovation is maintaining pointwise estimate of the arm's CDF, and pointwise lower confidence interval of the arm's CDF, which corresponds to the distribution of a random variable which stochastically dominates the given arm. This tweak makes the algorithm and its analysis easy to grasp, while still preserving connections to the lienar CMAB problem. In what follows, we adopt the notation of the present work: T is the horizon, m the number of arms, K the largest cardinality of a super-arm, M the max-reward of a set, alpha the approximation factor, C the lipschitz constant of the reward function. When the distributions are finitely supported, they present an upper bound whose asymptotic alpha-regret in the limit of the horizon T-> \infinity does not depend on s, the size of the largest support of one arms distribution. Moreover, by appropriately discretizing the algorithm for finitely supported distributions, they extend their result to an upper bound for generalized CMAB with arbitrary (possibly non-discrete) distributions, which does not require foreknowledge of the time horizon. Under the relatively mild assumption C / M = O(1), this upper bound matches the worst-case regret lower bounds for linear CMAB of O(sqrt(mKt)) up to a factor of sqrt(logT). This reviewer encourages the authors to more fully elaborate how their guarantees compare with known lower bounds in their dependence in the parameter K, m, and T, so future readers can appreciate the strength of their result: i.e, for general Lipschitz functions introduce no further worst-case statistical difficulty in the CMAB game than do linear functions. Paper 903 gives detailed attention to the K-MAX loss function, where the goal is to select the subset of arms which maximizes Exp[max_{i in S} ] over all subsets S of size K. The work presents a PTAS for a 1 - epsilon approximation oracle for every epsilon, yielding an online algorithm for obtaining vanishing 1-epsilon regret which runs in polynomial time. The construction and analysis of the PTAS are very clever and original even though (as is true of many PTAS's). While the run-time of the algorithm ( O(m^O((1/eps^2)))) is still prohibitive for any large m and moderate epsilon (for example, in light of the Gollovin and Streeter work, it only makes sense to take epsilon < 1/e, which yields a run time of a high degree polynomial in m), the authors do include experiments which verify that the PTAS is successful in yielding vanishing regret. It would have been interesting to see a comparison in run-time between the PTAS and brute-force search for small values of K, m, and small discretizations. Overall this is fantastic contribution: clearly written, original, and ostensibly correct. The only real weakness of the work is that the analysis is rather pessimistic, as the O(log-T) instance upper bounds only depend on the worst-case gaps between subsets, given by Delta_{min}. This gap dependence can be very, very loose. For example, consider the K-MAX reward function, decisision [m]-choose-K, and suppose all the arms are Bernoulli. In this special case, choosing the best subset of arms amounts to choosing those with the top-means. To make this concrete, suppose that there are K arms with mean mu + epsilon, and the remaining m-K arms have meaning mu. Then S^* is the top K arms,the gap between any arm i in S^* and j not in S^* is epsilon, but the gap in reward between S^* and any S \ne S^* is as small as Delta_{min} = (mu + epsilon)^K - (mu + epsilon)^{K-1}epsilon = epsilon(mu + epsilon)^{K-1}. Thus, the present works upper bound obtains a guarantee of the form O( m K log T/epsilon * 1/(mu + epsilon)^{K-1}). When mu + epsilon are bounded away from 1, 1/(mu + epsilon)^{K-1} = exp(-Omega(K)). However, there are algorithms for top-K identification see [Kalyanakrishnan, Shivaram, et al. "PAC subset selection in stochastic multi-armed bandits." Proceedings of the 29th International Conference on Machine Learning (ICML-12). 2012] finds that the best subset of arms can be identified with probability 1-delta using at most O(log(K/delta) m/epsilon^2) samples. If we consider each pull of a super arm as one "sample", then [Best-of-K Bandits, Max Simchowitz, Kevin Jamieson, Benjamin Recht, COLT, 2016]. or [Top Arm Identification in Multi-Armed Bandits with Batch Arm Pulls, Kwang-Sung Jun, Kevin Jamieson, Robert Nowak, Xiaojin Zhu, AISTATS, 2016. PDF] can identify the best arm using O(log(K/delta) m/ (K*epsilon^2)). These can be turned into a low-regret algorithm with regret of roughly O( m Delta_{max}/K epsilon^2 * log KT), where Delta_{max} = (mu+epsilon)^K - mu^K, which is no larger than K * epsilon; and hence, the run time of adapting top-K identification is no worse than O(mlog(KT)/epsilon). Hence, in the regime where mu + epsilon is bounded away from 1, we see that the CMAB algorithm in this worst has guarantees which are exponential-worse than using top-K. Obviously, there are sacrifices to be made for generality (indeed, the Bernoulli case is trivial and the innovation of this paper is that it works for non-Bernoulli distribution!). Moreover, I understand that even the linear CMAB upper bounds dont't have such a granular dependence on problem parameters. However, it would be nice to include further comments on the dependence on problem parameters, and potential looseness therein, in future revision. As an example of a highly-granular dependence on problem parameter, I recommend that you cite [Best-of-K Bandits, Max Simchowitz, Kevin Jamieson, Benjamin Recht, COLT, 2016]. The second half of that work provides upper bounds for the Bernoulli K-MAX problem under various partial feedback models (the first half is concerned with hardness results when the arms are not independent). While Paper 903's upper bound of O(mKlog T/Delta_min) can be exponentially (in K) loose for the semi-bandit feedback regime (as described above), the upper bounds and lower bounds in Simchowitz '16 for Bernoulli K-MAX under bandit feedback (only observing the reward of a super-arm, rather than the rewards of its constituents) do in fact incur a dependence on so called "information" occlusion terms, which are related to the gaps in rewards between super-arms. One other possibly area for improvement I seemed to find was that the algorithm for discrete distributions incurs an up-front (i.e., independent of T) regret of O(alpha * m * s), which seems quite hefty for short horizons. By adjusting the lower confidence bound in the line 12 of algorithm to look like O(sqrt(log (t * s)/T_i), it seems like one may be able to reduce this up-front regret to O(alpha * m) (i.e., no dependence on s), perhaps at the expense of a slight dependence on s in the asymptotic regret. I understand that the point was to have no dependence on s in the asymptote, but perhaps tweaking the confidence interval might allow for some nice interpolation. This may also yield improved analysis of the discretization for non-finitely supported distributions. ------- After Rebuttal period ------- Have reviewed authors comments and maintain my initial review.

Confidence in this Review

3-Expert (read the paper in detail, know the area, quite certain of my opinion)